# LANGUAGE GUIDED REPRESENTATION LEARNING

## ABSTRACT

Deep neural networks have achieved notable success; however, they still encounter significant challenges compared to humans, particularly in areas such as shortcut learning, texture bias, susceptibility to noise, and catastrophic forgetting, all of which hinder their ability to generalize and adapt. Humans excel in learning high-level abstractions, attributed to various mechanisms in the brain, including reasoning, explanation, and the ability to share concepts verbally—largely facilitated by natural language as a tool for abstraction and systematic generalization. Inspired by this, we investigate how language can be leveraged to guide representation learning. To this end, we explore two approaches to language guidance: Explicit Language Guidance, which introduces direct and verbalizable insights into the model, and Implicit Language Guidance, which provides more intuitive and indirect cues. Our extensive empirical analysis shows that, despite being trained exclusively on text, these methods provide supervision to vision encoders, resulting in improvements in generalization, robustness, and task adaptability in continual learning. These findings underscore the potential of language-guided learning to develop AI systems that can benefit from abstract, high-level concepts, similar to human cognitive abilities.

## 1 INTRODUCTION

Deep Neural Networks (DNNs) have demonstrated significant advancements in visual perception tasks and have surpassed test accuracy on many benchmark datasets. Despite their notable successes, there remains a considerable divide between the capabilities of DNNs and human intelligence. DNNs often struggle with out-of-distribution (OOD) data, rely on shortcut learning, exhibit texture bias, and are highly vulnerable to adversarial perturbations. Additionally, they face challenges when adapting to new data while maintaining previously learned knowledge in dynamic, non-stationary environments. In contrast, systematic generalization (Bahdanau et al., 2018),—the ability to compose and infer new meanings from previously learned concepts—is one of the aspects of human cognition that is still a challenge for neural networks and hampers their ability to generalize beyond the training distribution.

A common issue in DNNs is shortcut learning (Geirhos et al., 2020; Jo & Bengio, 2017), where models rely on spurious correlations or superficial features in the data rather than learning the true underlying causal patterns. For instance, a model trained to recognize birds might associate specific backgrounds, such as the sky or trees, with bird species, rather than focusing on the salient features of the bird itself. Similarly, neural networks often exhibit texture bias, focusing on local textures (Geirhos et al.), rather than semantic features. These reliances lead to poor generalization, particularly when the model encounters new, unseen data where these shortcuts or textures do not apply. Moreover, DNNs lack robustness in the face of adversarial perturbations—small, often imperceptible changes to input data that can drastically alter a model's predictions. While humans are largely unaffected by such minor variations, these perturbations remain a significant vulnerability for DNNs, highlighting a huge limitation in safety-critical applications.

In addition to these, DNNs face significant challenges in the context of continual learning (Parisi et al., 2019). Many real-world applications involve non-static, sequential data, where models are exposed to a potentially endless stream of tasks, requiring them to learn incrementally over time. Unlike humans, who can relatively learn new tasks while retaining previously acquired knowledge to a better extent, DNNs suffer from catastrophic forgetting. When trained on new tasks sequentially, DNNs often overwrite earlier representations, causing a dramatic decline in performance on previ-

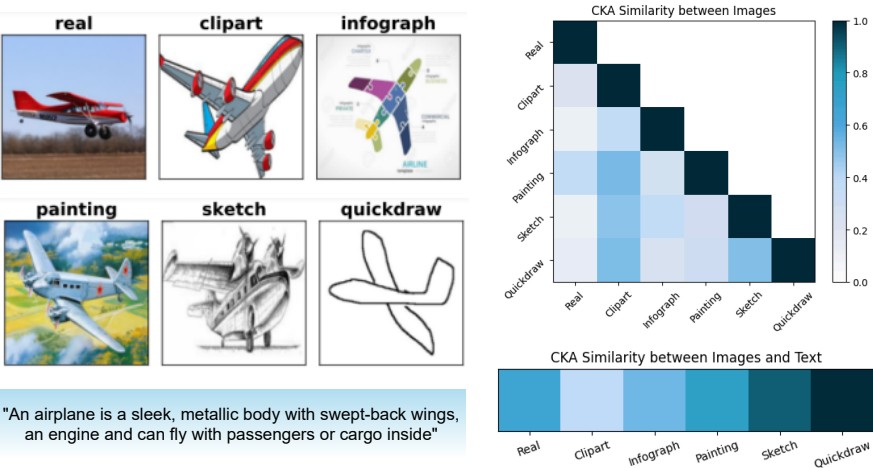

Figure 1: Feature similarity between images of different domains and between image and language. Even in challenging domains, text modality provides shared semantic concepts that can enhance model generalization.

ously learned tasks. This issue is particularly challenging in dynamic environments where models must continuously adapt to new information. The inability of DNNs to balance the incorporation of new knowledge while preserving prior learning hinders their development for lifelong learning. Addressing these limitations is crucial for developing neural networks capable of functioning effectively in real-world environments, that are dynamic and continuously evolving.

## 2 INDUCTIVE BIAS

To bridge the gap between neural networks and the cognitive competence displayed by humans, we revisit the concept of inductive biases. According to the no-free-lunch theorem for machine learning (Wolpert et al., 1995) achieving generalization requires a set of preferences or assumptions over the space of all possible functions. Inductive bias refers to these underlying assumptions that guide a learning algorithm toward specific types of solutions, enabling it to generalize beyond the finite set of training data. In the case of DNNs, inductive biases can manifest as structural or high-level priors, or even as auxiliary knowledge. Humans learn high-level abstractions and this ability is attributed to various mechanisms in the brain and is often facilitated by language, which allows these abstractions to be verbalized (Goyal & Bengio, 2022). These abstractions, grounded in language, aid in systematic generalization by allowing them to reason, imagine, and explain at an explicit, language-driven level. This ability to infer abstract concepts—such as causal relationships and object interactions—plays a critical role in their ability to generalize across different contexts. Incorporating similar priors into DNNs could improve their capacity for abstraction, and generalization across diverse and novel scenarios.

An additional intriguing aspect is how these high-level representations are shared and integrated in the brain. Cognitive theories provide insights into this, particularly the distinction between System 1 (Implicit) and System 2 (Explicit) processing (Kahneman, 2011). There is explicit (verbalizable) knowledge and explicit processing in system2, and implicit (intuitive) knowledge in system1 (Goyal & Bengio, 2022). Explicit knowledge is consciously accessible and can be reasoned and shared through language. Implicit knowledge refers to intuitive understandings that are difficult to articulate. Another relevant theory is the Global Workspace Theory (GWT (Baars, 1993; Dehaene & Naccache, 2001), which offers a framework for understanding how specialized modules in the brain communicate through a shared cognitive workspace (Juliani et al., 2022). This workspace allows information to be broadcast across different regions, enabling alignment and collaboration among various processes. The GWT posits that this shared communication framework facilitates the integration of semantic knowledge across modalities, allowing for the formation of more abstract and

high-level representations. This can result in semantic knowledge that is not tied to a specific modality and is more generic and rich in high-level abstract concepts.

## 3 ROLE OF LANGUAGE

Inspired by these insights, exploring natural language and the ways it can be integrated effectively into vision-based learning, becomes a compelling avenue for research. We hypothesize that language can add guidance to vision-based training to create richer, more semantically meaningful representations of visual data. This approach allows the model to leverage linguistic knowledge to fill in gaps in visual information, leading to more accurate and contextually relevant outputs.

The integration of language into visual representation learning taps into the shared semantic space that both modalities occupy. An example is highlighted in Figure 1, where we take an image of the same object (an airplane) in varying domains, some more challenging than others. The first heatmap shows the Central Kernel Alignment (CKA) similarity between images from different domains. Darker shades indicate higher similarity between the features. The second heatmap measures the CKA similarity between the visual representations of images and the generic text description (of what an airplane looks like). As shown in the similaity matrix, challenging image types, like infographs and paintings, are more difficult to adapt to, when using visual features alone. However, they seem to map more closely in text-based representations, as the semantic content carries more information than purely visual features. This emphasizes how language models provide an abstract, conceptual understanding that transcends surface-level visual similarities and aids in learning shared representations for improved generalization across different visual domains.

## 4 LANGUAGE GUIDANCE IN REPRESENTATIONAL LEARNING

In this work, we seek to explore how language can be used as a tool to guide representation learning. We hypothesize that utilizing both visual features (textural and low-level information), alongside high-level abstractions derived from language, can help produce semantically rich representations, that can aid in different forms of generalization. Our work investigates several key questions:

- Can language be used to guide representational learning?
- How can we leverage pre-trained language models to produce rich representations in the visual domain?
- Can language models, only having seen language, generalize to visual perception tasks?

We aim to utilize pre-trained language models in various ways to offer guidance and improve the training process in conventional vision-based supervised learning. Foundation models, also known as pre-trained models, constitute a pivotal aspect of contemporary AI research. These models are trained on vast amounts of data, enabling them to generalize effectively across a wide range of downstream tasks. Sentence Transformers (Reimers & Gurevych, 2019) and models like LLAMA (Large Language Model Meta AI) (Touvron et al., 2023) are a few powerful language models trained on extensive text corpora, designed to produce high-quality semantic representations. While these models have been used across a variety of NLP tasks, they also present opportunities for application in the visual domain. We investigate how frozen language models—without further fine-tuning or additional training—can be used to guide the training of vision encoders.

In contrast to Vision-Language Models (VLMs) (Desai & Johnson, 2021; Radford et al., 2021; Alayrac et al., 2022), which jointly train encoders on vision and text data for tasks like Visual Question Answering (VQA) (Antol et al., 2015) and image captioning, our exploration takes a different direction. VLMs typically align visual and textual information within a shared embedding space, requiring multi-modal datasets and the joint training of both vision and language encoders. Rather than delving into the domain of training multiple encoders, using multi-modal data, fine-tuning, or employing prompt-based learning, we aim to investigate a more fundamental question of how the knowledge embedded in the language model can be leveraged to influence vision encoder training. We aim to examine whether this simple transfer of knowledge from language to vision offers any advantages.

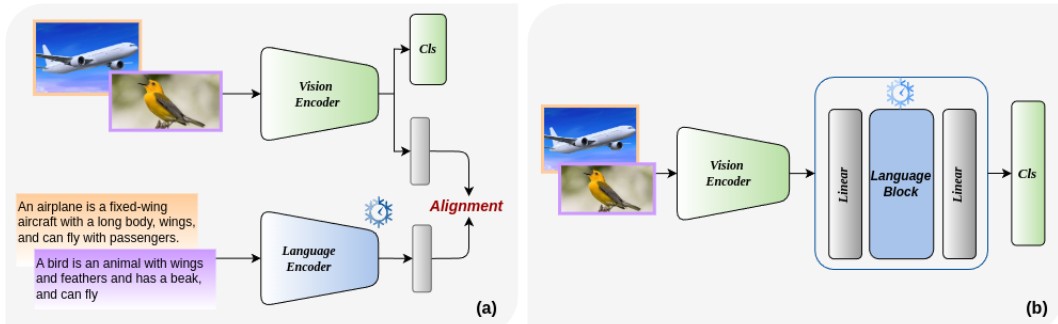

Figure 2: (a) Explicit Language Guidance: Learning visual representations with explicit supervision from language descriptions. (b) Implicit Language Guidance: Learning visual representations via an embedded frozen language block for implicit supervision.

Building on the Explicit-Implicit theory, we investigate two key approaches for incorporating language into visual learning: (1) Explicit Language Guidance, where language descriptions play a direct explicit role in shaping the learning process, and (2) Implicit Language Guidance, where pre-trained language model indirectly supports the learning.

### 4.1 EXPLICIT GUIDANCE: LEARNING VISUAL REPRESENTATIONS WITH EXPLICIT KNOWLEDGE ALIGNMENT FROM LANGUAGE

In Explicit Language Guidance (ExLG), we utilize explicit information such as language descriptions of the objects to guide the training process. The approach uses a typical vision encoder to process image data and a classifier for decision-making. The high-level semantic descriptions of the objects are introduced via a pre-trained language model. This model provides rich language-based embeddings from descriptions, generated either manually or using models like GPT (Achiam et al., 2023). While the language encoder remains frozen during training (i.e., its parameters are not updated), its embeddings are used to guide the vision encoder (Figure 2).

To leverage both visual and textual information, we align the representations from the vision and language encoders. A similarity-preserving loss (Tung & Mori, 2019) guides the vision encoder by ensuring that input pairs with similar activations in the language model also produce similar activations in the vision model. Specifically, the similarity-preserving loss works by computing pairwise similarity matrices from the activation maps of both the vision and language models. The loss function penalizes differences between these similarity matrices, encouraging the vision model to learn representations that are aligned with the semantic knowledge embedded in the language descriptions.

The overall loss function is defined as:

$$\mathcal{L} = \mathcal{L}_{\text{cls}} + \lambda \mathcal{L}_{\text{align}} \tag{1}$$

where $\mathcal{L}_{\text{cls}}$ is the classification loss, $\mathcal{L}_{\text{align}}$ is the alignment loss, and $\lambda$ controls the influence of the alignment term. The alignment loss is defined as:

To guide the vision encoder towards the activation correlations induced in the language encoder, we define a similarity-preserving distillation loss :

$$\mathcal{L}_{\text{align}} = \frac{1}{N^2} \|\mathcal{S}_v - \mathcal{S}_l\|^2 \tag{2}$$

$$\mathcal{S}_v = \frac{f_v \cdot f_v^\top}{\|f_v\| \cdot \|f_v\|}, \quad \mathcal{S}_l = \frac{f_l \cdot f_l^\top}{\|f_l\| \cdot \|f_l\|} \tag{3}$$

$f_v$ and $f_l$ are the feature matrices for vision and language encoders at the chosen layer. The goal of the similarity-preserving loss is to align the similarity structure in the vision embedding space with that of the language embedding space.

## 4.2 Implicit Guidance: Learning Visual Representations via an Embedded Language Encoder

The implicit approach involves integrating a pre-trained Large Language Model (LLM) directly within the vision encoder, with the goal of getting indirect supervision, without requiring any other data (Figure 2). This approach is inspired by recent works investigating the potential for language models (LMs) to generalize beyond linguistic tasks. Research has shown that text transformers, even when trained exclusively on text data, can develop multi-modal neurons—neurons that respond similarly to both image and text embeddings with semantically related meanings (Schwettmann et al., 2023). Additionally, studies (Pang et al., 2023) have shown the versatility of using LLM blocks for vision encoders and their ability to act as a filter, amplifying relevant features and distilling important information from visual inputs. Building on these insights, our goal is to explore whether LLMs can enhance generalization, robustness, and continual learning in vision tasks. Specifically, we seek to determine if LLMs can act as a source of implicit textual knowledge, directing attention toward more informative visual features and mitigating challenges like catastrophic forgetting.

In our study, we implement this approach by adding a frozen language encoder block after the vision encoder. To ensure dimensional compatibility, we introduce linear layers to map the vision encoder's features to the input dimensions required by the language model block. Classification is performed on these transformed features without incorporating any additional loss functions or regularization.

## 5 Empirical Study

In this section, we comprehensively evaluate the performance of language-guided models across a range of scenarios, using multiple datasets. We begin by exploring IID generalization, followed by an analysis of OOD performance. Further, we evaluate scenarios involving shortcut learning and texture bias. We also test the robustness of models against adversarial attacks. To further assess the applicability of language guidance, we extend our evaluation to continual learning benchmarks, aiming to understand its effectiveness in mitigating catastrophic forgetting. For our experiments, we use a ResNet-18 (He et al., 2016) architecture as the vision encoder and a Sentence Transformer (Reimers & Gurevych, 2019) as the language encoder. Our analysis spans several datasets, including CIFAR10, CIFAR100, TinyImageNet, and various forms of ImageNet for OOD. Additionally, we incorporate Tinted-CIFAR and Skewed-CelebA to examine shortcut learning scenarios, and standard continual learning datasets for evaluating continual learning performance. Detailed experimental setups and additional architectures are provided in the Appendix.

### 5.1 IID and OOD Generalization

In supervised learning, Independent and Identically Distributed (IID) generalization refers to the model's ability to maintain performance on test data that follows the same distribution as the training data. In contrast, Out-of-Distribution (OOD) generalization evaluates how well a model performs when presented with data that deviates from the training distribution, an essential criterion for robust machine learning models deployed in real-world scenarios. We benchmark three models: the Baseline model, a conventional classification model comprising a vision encoder network paired with a classifier, trained on an image dataset using supervised learning with a cross-entropy loss. Alongside this, we evaluate two variants that incorporate language guidance—ExLG (Explicit Language Guidance) and ImLG (Implicit Language Guidance).

We test these models on standard datasets such as CIFAR-10, CIFAR-100, and TinyImageNet. Additionally, we explore the sample efficiency of each model by gradually reducing the training data and examining how well the models retain performance with less data, simulating low-data regimes. For the OOD evaluation, we assess the models' robustness on challenging benchmarks derived from the ImageNet dataset, namely ImageNet-O (which contains outlier data points that lie outside the training classes), ImageNet-R (comprising artistic renditions of objects), and ImageNet-A (which contains adversarially filtered images known to challenge standard models) (Hendrycks et al., 2021a;b).

Table 1 shows that ExLG model consistently performs better in IID settings across all datasets. It also shows superior results in low-data scenarios, further highlighting the benefit of explicit supervision from language models. Notably, ExLG outperforms ImLG in these settings, indicating that direct language supervision provides more utility in in-distribution testing. The ImLG model

Table 1: IID and OOD generalization across various datasets, with sample efficiency evaluated on limited CIFAR10 data. ExLG excels overall, while ImLG shows strong performance in OOD scenarios.

| Method | CIFAR-10 | Sample Efficiency | | | | |
|---|---|---|---|---|---|---|
| | | 2% | 5% | 10% | 20% | 50% |
| Baseline | $94.84_{\pm0.14}$ | $45.71_{\pm1.52}$ | $55.42_{\pm1.08}$ | $67.04_{\pm2.19}$ | $79.62_{\pm2.60}$ | $90.08_{\pm1.80}$ |
| ExLG | $\mathbf{95.12}_{\pm\mathbf{0.05}}$ | $\mathbf{47.88}_{\pm\mathbf{0.53}}$ | $\mathbf{57.24}_{\pm\mathbf{1.95}}$ | $\mathbf{69.97}_{\pm\mathbf{1.87}}$ | $\mathbf{84.75}_{\pm\mathbf{0.61}}$ | $\mathbf{92.26}_{\pm\mathbf{0.06}}$ |
| ImLG | $93.41_{\pm0.46}$ | $45.03_{\pm2.06}$ | $55.53_{\pm2.06}$ | $67.82_{\pm1.42}$ | $79.03_{\pm0.57}$ | $89.40_{\pm0.37}$ |

| Method | CIFAR-100 | TinyImageNet | ImageNet100 | OOD Generalization | | |
|---|---|---|---|---|---|---|
| | | | | ImageNet-O | ImageNet-R | ImageNet-A |
| Baseline | $76.98_{\pm0.39}$ | $58.73_{\pm0.35}$ | 71.46 | $41.73_{\pm1.45}$ | $10.59_{\pm0.41}$ | $1.92_{\pm0.53}$ |
| ExLG | $\mathbf{77.59}_{\pm\mathbf{0.08}}$ | $\mathbf{65.63}_{\pm\mathbf{0.26}}$ | **79.42** | $\mathbf{46.70}_{\pm\mathbf{1.02}}$ | $\mathbf{14.95}_{\pm\mathbf{0.07}}$ | $\mathbf{2.94}_{\pm\mathbf{0.33}}$ |
| ImLG | $74.10_{\pm0.91}$ | $60.02_{\pm0.16}$ | 72.37 | $42.20_{\pm0.81}$ | $12.10_{\pm0.17}$ | $2.37_{\pm0.27}$ |

Table 2: Shortcut learning on Tinted-CIFAR10 and Skewed-CelebA dataset. Language-guided models are less vulnerable to the spurious features added to the dataset.

| Method | Tinted-CIFAR10 | Skewed-CelebA | | | | |
|---|---|---|---|---|---|---|
| | | Final | NonBlonde-M | Blonde-F | Blond-M | NonBlonde-F |
| Baseline | $16.45_{\pm1.81}$ | $61.28_{\pm1.21}$ | $94.71_{\pm0.08}$ | $92.21_{\pm1.02}$ | $56.38_{\pm0.39}$ | $27.74_{\pm2.29}$ |
| ExLG | $\mathbf{18.24}_{\pm\mathbf{0.60}}$ | $\mathbf{72.11}_{\pm\mathbf{1.28}}$ | $\mathbf{96.29}_{\pm\mathbf{0.30}}$ | $\mathbf{95.18}_{\pm\mathbf{0.31}}$ | $\mathbf{68.33}_{\pm\mathbf{0.98}}$ | $\mathbf{47.67}_{\pm\mathbf{2.31}}$ |
| ImLG | $\mathbf{18.51}_{\pm\mathbf{1.04}}$ | $\mathbf{75.90}_{\pm\mathbf{1.79}}$ | $\mathbf{97.85}_{\pm\mathbf{0.65}}$ | $\mathbf{96.81}_{\pm\mathbf{0.11}}$ | $\mathbf{69.77}_{\pm\mathbf{1.03}}$ | $\mathbf{53.84}_{\pm\mathbf{3.38}}$ |

performs better than the baseline in OOD settings, particularly on ImageNet-O, ImageNet-R, and ImageNet-A. The frozen language model used in ImLG helps the vision encoder by filtering and amplifying important visual features, allowing the model to focus on relevant regions, thus improving generalization to other distributions.

## 5.2 SHORTCUT LEARNING

Shortcut learning is a common problem in neural networks, where models rely on superficial patterns or spurious correlations present in the training data to make predictions, rather than learning meaningful representations (Geirhos et al., 2020). This behavior leads to poor generalization, especially when models are evaluated on data that differs from their training distribution. To test the extent of shortcut learning, we employ two specially curated datasets: Tinted-CIFAR10 and Skewed-CelebA. Tinted-CIFAR10: In this variant of CIFAR10, a unique color tint is added to each class. This dataset tests whether the models use the color tint as a spurious cue for classification. Skewed-CelebA: In this skewed version of the CelebA (Liu et al., 2015) dataset, the training data is heavily biased. It consists primarily of blonde women and non-blonde men. During evaluation, however, the models are tested on non-blonde women and blonde men—categories they have never seen during training.

As seems in Table 2, the baseline model performs poorly across both datasets. Language guidance improves performance over the baseline, particularly on Skewed-CelebA. With explicit language guidance, the model significantly improves its performance. In particular, ExLG shows a 22% improvement for blonde males and a 70% improvement for non-blonde females, the categories never seen in training distribution. The improvement is even higher in the implicit language-guided model, boosting overall accuracy from 61.28 to 75.90, and a massive 94% improvement in the non-blonde female category. To further get insights into this behavior, we use Grad-CAM (Selvaraju et al., 2017) to generate activation maps on the Skewed-CelebA dataset. These maps, shown in Figure 3, reveal how the models focus on different parts of the image. The baseline model predominantly focuses on superficial cues like hair color or background. In contrast, the ExLG and ImLG models, trained with language guidance, focus on more salient facial features to make decisions.

ImLG, in particular, outperforms ExLG because it leverages the frozen language model's ability to act as a conceptual filter. This filter enhances the model's focus on high-level, task-relevant infor-

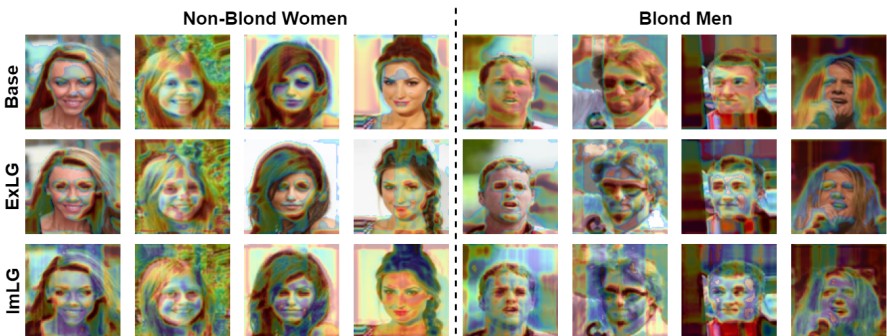

Figure 3: Activations maps of the models on the Skewed-CelebA dataset. Language-guided models focus on the salient features, while conventional methods focus on spurious cues (hair color).

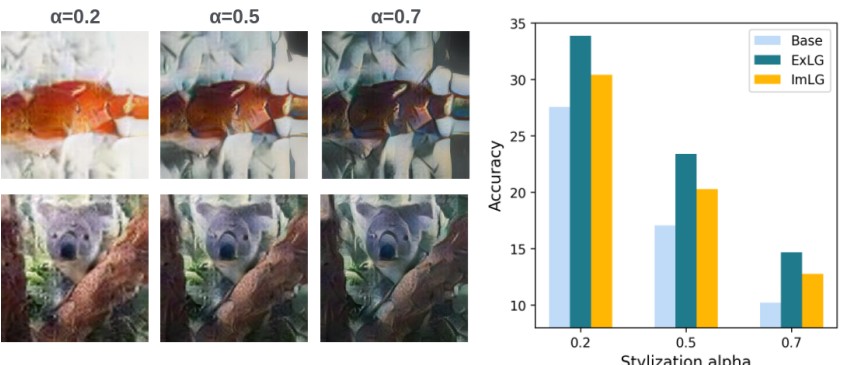

Figure 4: Analysis on Stylized TinyImageNet across three levels of stylization. Language-guided models demonstrate better generalization, reducing texture bias.

mation while disregarding superficial patterns, such as textures or spurious correlations. The results suggest that incorporating language into the models enables them to develop a deeper semantic understanding of the underlying concepts in the images, allowing for stronger performance even in challenging test scenarios.

### 5.3 TEXTURE BIAS

Deep neural networks often rely heavily on texture information when making predictions (Geirhos et al.). This reliance on texture can lead to a bias, and limit the model's ability to generalize to more diverse or out-of-distribution data. To evaluate texture bias and investigate the extent to which models rely on texture cues, we perform style transfer (Huang & Belongie, 2017) on the TinyImageNet dataset. By applying style transfer, we generate stylized images with various texture patterns, while keeping the underlying object shapes intact. The stylization alpha determines the extent to which the original image's texture is replaced with the style features from a reference image. We use three different levels to progressively increase the degree of texture variation in the images.

Figure 4 shows some sample images and also the performance graph. The baseline model, which relies more heavily on local texture information, experiences a significant drop in accuracy as the stylization increases. In contrast, both language-guided (LG) models, ExLG and ImLG, perform better across all stylization levels compared to the baseline model. The LG models' superior performance suggests that these models are able to learn more abstract and global representations of the data, allowing them to better generalize in the presence of significant texture changes.

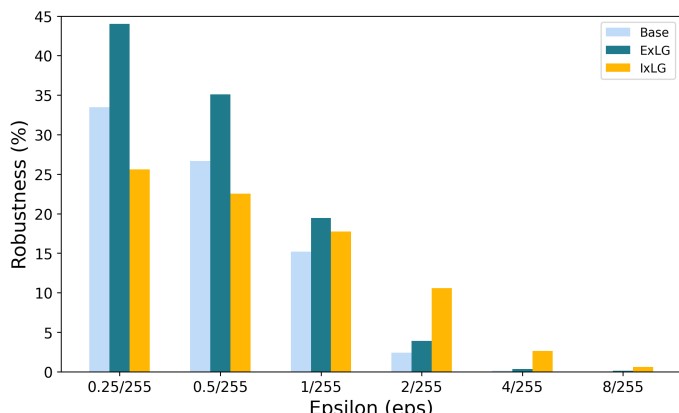

Figure 5: Robustness analysis to PGD-10 adversarial attack on varying strengths ($\epsilon$) on CIFAR10 dataset.

Table 3: Effect of language guidance to class-incremental learning on multiple datasets with varying buffer sizes.

| Buffer | Method | Seq-CIFAR10 | Seq-TinyImageNet | DN4IL |
|---|---|---|---|---|
| - | SGD | $19.62_{\pm0.05}$ | $7.92_{\pm0.26}$ | $20.83_{\pm0.24}$ |
| - | Joint | $92.20_{\pm0.15}$ | $59.99_{\pm0.19}$ | $59.93_{\pm1.07}$ |
| 200 | ER | $44.79_{\pm1.86}$ | $18.38_{\pm0.16}$ | $24.15_{\pm0.34}$ |
| | ExLG | $\mathbf{54.84_{\pm0.97}}$ | $\mathbf{20.39_{\pm0.15}}$ | $\mathbf{27.71_{\pm0.64}}$ |
| | ImLG | $\mathbf{47.57_{\pm0.20}}$ | $\mathbf{19.86_{\pm0.24}}$ | $\mathbf{24.22_{\pm0.12}}$ |
| 500 | ER | $57.74_{\pm0.27}$ | $19.85_{\pm0.39}$ | $30.96_{\pm0.62}$ |
| | ExLG | $\mathbf{67.03_{\pm0.21}}$ | $\mathbf{21.68_{\pm0.20}}$ | $\mathbf{31.67_{\pm0.32}}$ |
| | ImLG | $\mathbf{62.49_{\pm0.99}}$ | $19.57_{\pm0.45}$ | $29.98_{\pm0.85}$ |

## 5.4 ROBUSTNESS

DNNs, though highly effective at learning patterns in data, are notably vulnerable to adversarial attacks (Szegedy, 2013). Adversarial attacks are small, imperceptible perturbations to the input that can cause significant changes in the model's output. In comparison to humans, who are generally resistant to such subtle manipulations in images, DNNs can be easily fooled, making them susceptible to real-world attacks. In this section, we evaluate the adversarial robustness of the models using Projected Gradient Descent (PGD) attacks (Madry, 2017) on the CIFAR-10 dataset. PGD is a powerful iterative attack method that perturbs the input image in small steps to fool the model by progressively maximizing the model's loss. To test the robustness of our models, we apply attacks with increasing strengths, measured by the perturbation magnitude $\epsilon$, and assess the models' ability to maintain performance in the face of these adversarial examples.

As shown in Figure 5, the LG-Ex model consistently surpasses the baseline model (Base-Cls) across all levels of attack strength, demonstrating stronger adversarial robustness. The interesting observation, however, comes from the behavior of the ImLG model. While its performance is lower than both ExLG and Baseline at lower attack strengths, it becomes significantly more robust as the attack strength increases. For higher attack magnitudes, ImLG outperforms both ExLG and Baseline, showcasing superior resilience to stronger attacks. Jo & Bengio (2017) hypothesize that if models are truly learning high-level abstractions, they should be resilient to perturbations in the data. Therefore, the integration of language guidance not only enhances task performance but also facilitates the development of more robust representations.

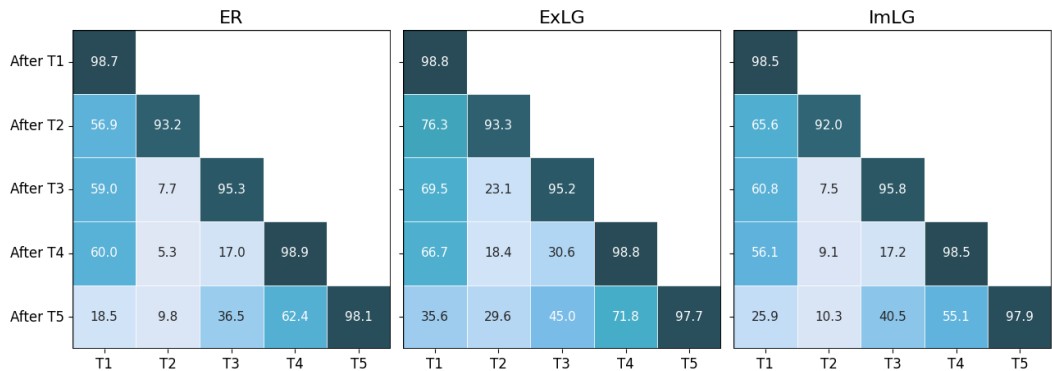

Figure 6: Task-wise performance of class-incremental learning setting on Seq-CIFAR10 dataset with 200 buffer size.

## 5.5 CONTINUAL LEARNING

Continual Learning (CL) (Parisi et al., 2019) focuses on the challenge of learning new tasks sequentially without forgetting previously learned tasks, a phenomenon known as catastrophic forgetting. Within CL, class-incremental learning (Class-IL) is particularly difficult, as new classes are introduced over time, and the model must not only adapt to these new classes but also retain its knowledge of earlier classes. Another paradigm is domain-incremental learning (Domain-IL), where the same classes are presented across different domains, adding the challenge of domain shifts. Both settings test the model's ability to generalize and prevent forgetting (Van de Ven & Tolias, 2019).

In our experiments, we employ the standard replay method, Experience Replay (ER) as the baseline, which mitigates forgetting by replaying samples from a fixed buffer (Buzzega et al., 2020). In Table 3, we present results in the Class-IL setting across two datasets: Seq-CIFAR-10 and Seq-TinyImageNet. In the 200-buffer setting, ExLG outperforms the baseline on both buffer sizes. Figure 6 shows the task-wise performance after each task.

The last row specifically shows the accuracy on all the tasks after the model finishes learning the final task. For example, the accuracy on Task 1 drops from 98.7 to 18.5 by the time the model finishes learning Task 5. The ExLG and ImLG models demonstrate better performance across all the tasks. The performance drop in old tasks is much lower compared to the baseline. This demonstrates that supervision to vision model is more effective at mitigating catastrophic forgetting, preserving more of the knowledge from earlier tasks as it learns new ones. The use of language guidance helps the model learn semantic shared concepts that can be present in many tasks, thereby achieving better long-term retention and domain adaptation throughout the incremental learning process.

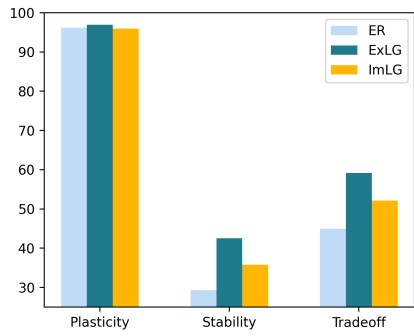

Figure 7: Plasticity-stability trade-off analysis on Seq-CIFAR10 dataset with 200 buffer size.

**Plasticity-Stability Trade-off** refers to the balance between a model's ability to learn new tasks (plasticity) and its ability to retain knowledge from previous tasks (stability). In continual learning settings, models often struggle to maintain this balance. In Figure 7, the Base model shows high plasticity, meaning it excels at learning new tasks. However, this comes at the cost of low stability, as seen in the stability metric, indicating that it forgets much of the old information when learning new tasks. In contrast, the language-guided models (ExLG and ImLG) exhibit much higher stability. This suggests that these models are better at retaining previously learned information, while still adapting to new tasks.

Table 4: Analysis of the effect of different language modules on the implicit language-guided model: Classification performance on CIFAR10 with ViT-Tiny as the vision encoder.

| | Vision Only | + Implicit Language Guidance | | |
|---|---|---|---|---|
| | | LLAMA-8B | CLIP | Sentence Transformer |
| CIFAR10 | 93.38 | **94.19** | **94.03** | 93.12 |

## 6 Architecture Analysis

In this section, we analyze how different architectures impact the performance of language guidance. In our earlier ExLG experiments, we replaced the Sentence Transformer with a CLIP text encoder (Radford et al., 2021), observing comparable performance across both models. Unlike the explicit approach, the implicit method integrates a transformer block directly into the vision encoder, implicitly providing supervision through its attention mechanisms. To further explore how attention operates when both the vision and language components are transformer-based, we replaced the traditional CNN vision encoder with a transformer-based architecture, specifically ViT-Tiny. Moreover, we scaled our investigation by integrating larger language models, including LLAMA (Dubey et al., 2024) and CLIP, with the latter being pretrained on multimodal data.

As shown in Table 4, the IID performance remains comparable to the baseline, though LLAMA shows the highest performance among the LM models. In Figure 8, we illustrate the impact of the language block through activation maps. Despite using a transformer-based vision encoder, the LM block still effectively guides the model to focus on more salient and semantically relevant regions of the image. The Baseline model (left) shows limited focus, often missing key regions in the images. In contrast, the Implicit method (right) integrates a frozen LM, which helps the vision encoder concentrate on task-relevant visual features. Overall, these results reinforce the idea that frozen LMs can enhance vision models by embedding abstract, transferable concepts, even when trained solely on textual data.

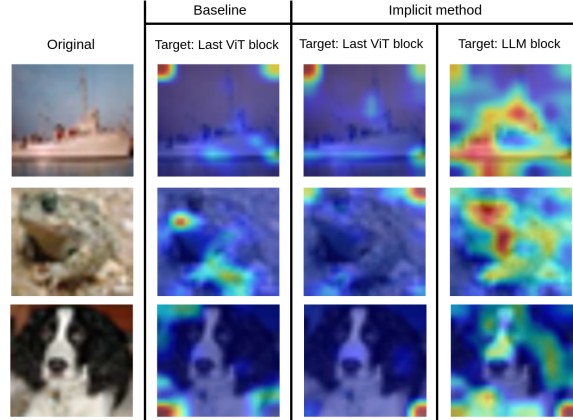

Figure 8: Activation maps after each block of implicit language-guided training.

## 7 Conclusion

We investigate how language can be leveraged for representational learning in vision models. We explore different strategies for leveraging the rich information embedded in pre-trained language models to create more semantic and robust representations. The explicit approach integrates language directly into the training process by aligning visual and textual representations while the implicit approach offers indirect guidance from language. The explicit approach performs better overall in various generalization tasks and continual learning. On the other hand, the implicit approach shows better performance in challenging scenarios, particularly in shortcut learning, texture bias analysis, and under severe adversarial attacks. The same advantages of the implicit approach do not fully translate to test accuracy (in-distribution performance), likely due to the lack of alignment between the vision and language encoders, highlighting the potential for further exploration and optimization of the implicit method. Overall, this work underscores the potential of language-guided learning to build more robust, adaptable, and semantically rich representations in vision tasks, offering promising pathways for improving generalization and resilience in AI models.

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

## A EXPERIMENTATION SETTING

The summary of all the extensive analysis in the paper along with the corresponding datasets is shown in Table 5.

Table 5: Summary of all analyses and datasets in this study

| Vision Enc
Language Enc | ResNet18
Sentence Transformer variants | ResNet50
CLIP | VIT
CodeBERT | LLAMA |
|---|---|---|---|---|
| Analysis | Datasets | | | |
| IID | CIFAR10 | CIFAR100 | TinyImageNet | ImageNet100 |
| OOD | ImageNet-O | ImageNet-R | ImageNet-A | |
| Shortcut Learning | Tinted-CIFAR10 | Skewed-CelebA | | |
| Texture Bias | Stylized TinyImageNet | | | |
| Adversarial Robustness | CIFAR10 | | | |
| Continual Learning | Seq-CIFAR10 | Seq-TinyImageNet | DN4IL | |

### A.1 IID AND OOD GENERALIZATION

For the IID (Independent and Identically Distributed) setting, we evaluate on CIFAR-10, CIFAR-100, and TinyImageNet, which are standard datasets used in classification tasks. To explore shortcut learning, we employ Tinted-CIFAR10 and Skewed-CelebA datasets, which introduce biases and distribution shifts designed to test the model's ability to avoid learning spurious correlations.

We conduct out-of-distribution (OOD) tests by training the model on TinyImageNet and testing it on three challenging OOD datasets: ImageNet-A, ImageNet-O, and ImageNet-R. ImageNet-A (Adversarial) (Hendrycks et al., 2021b) consists of naturally occurring adversarial examples that are misclassified by models trained on ImageNet, making it an ideal dataset for evaluating a model's adversarial robustness. ImageNet-O (Outliers) (Hendrycks et al., 2021b) contains outlier images that do not belong to any of the ImageNet classes, allowing us to test the model's ability to handle inputs outside of its training distribution. Lastly, ImageNet-R (Renditions) (Hendrycks et al., 2021a) includes artistic renditions of ImageNet classes, such as paintings, cartoons, and sculptures, which introduce significant style variations and help in evaluating the model's capacity for generalization across different visual domains.

### A.2 CONTINUAL LEARNING

In the continual learning setting, we explore Class-Incremental Learning (Class-IL) and Domain-Incremental Learning (Domain-IL), both of which are common benchmarks for evaluating continual learning models. In Class-IL, each task introduces new classes, and the model is required to learn these new classes while retaining knowledge of previously learned classes without forgetting. In contrast, Domain-IL involves tasks where the class labels remain the same across tasks, but the input data distribution shifts with each new task. For Domain-IL, we focus on the DN4IL dataset.

The DN4IL (DomainNet for Domain-IL) dataset (Gowda et al., 2023) is a curated subset of the DomainNet dataset (Peng et al., 2019), originally used for domain adaptation tasks. It has common objects across six diverse domains: real, clipart, infograph, painting, quickdraw, and sketch DN4IL offers a more succinct, balanced, and computationally efficient version of DomainNet, making it well-suited for benchmarking continual learning methods while preserving the challenging distribution shifts between domains.

Plasticity measures the model's capability to learn new tasks. It is calculated as the average accuracy of each task when it is first learned. For example, this is the accuracy of the network trained on task $T_2$, evaluated on the test set of $T_2$. Stability measures the model's ability to retain knowledge from previously learned tasks. It is computed as the average accuracy of all tasks from 1 to $T - 1$ after

Table 6: Hyperparameters: All models are trained for 100 epochs using the SGD optimizer, except for implicit methods, which employ the AdamW optimizer due to the inclusion of a transformer block.

| Method | CIFAR10, CIFAR100, Tinted-CIFAR10 | Skewed-CelebA | TinyImageNet |
|---|---|---|---|
| ExLG | $lr = 0.1$ 
 $\lambda = 15.0$ | $lr = 0.03$ 
 $\lambda = 15.0$ | $lr = 0.03$ 
 $\lambda = 100.0$ |
| ImLG | $lr = 0.003$ | $lr = 0.0001$ | $lr = 0.003$ |

learning the final task $T$. Trade-off - To assess the balance between plasticity and stability, we use the following metric:

$$\text{Trade-off} = \frac{2 \times \text{Plasticity} \times \text{Stability}}{\text{Plasticity} + \text{Stability}}$$

### A.3 HYPER-PARAMETERS

For all the baseline experiments, we adopt standard classification settings. For CIFAR-10, CIFAR-100, Tinted-CIFAR10, and Skewed-CelebA, we use a learning rate of $0.1$ with SGD as the optimizer, training for 100 epochs. For TinyImageNet, we use a learning rate of $0.03$ while keeping the same number of epochs and optimizer settings.

There is only hyper-parameter for ExLG ($\lambda$) and no additional parameters for ImLG. The hyper-parameters for Explicit and Implicit methods are provided in The Tables 6 and 7. In the case of Implicit Language Guidance, we extract only the last block in every language model. We use the Adam optimizer with a weight decay of $5e - 4$.

## B CKA

Centered Kernel Alignment (CKA) is a widely used method for measuring the similarity between two representations in neural networks. It quantifies how well these representations align, allowing us to compare features learned by different layers or models. CKA is computed using dot products of representations in the form of Gram matrices, which capture pairwise similarities between examples in each representation. By comparing these Gram matrices, CKA evaluates the structural alignment between two sets of representations, making it particularly useful for understanding the alignment between the activations of a vision encoder and a language model.

The CKA similarity between two feature matrices, $X \in \mathbb{R}^{n \times d_1}$ and $Y \in \mathbb{R}^{n \times d_2}$, where $n$ is the number of samples and $d_1$ and $d_2$ are the dimensions of the features, is calculated as follows. First, we compute the centered Gram matrices $K$ and $L$ for $X$ and $Y$, respectively:

$$\text{CKA}(X, Y) = \frac{\text{HSIC}(K, L)}{\sqrt{\text{HSIC}(K, K) \cdot \text{HSIC}(L, L)}} \tag{4}$$

Here, HSIC (Hilbert-Schmidt Independence Criterion) measures the similarity between the two Gram matrices:

CKA is invariant to orthogonal transformations and isotropic scaling of the representations, making it a robust tool for comparing representations between models. By using CKA, we can effectively evaluate how well representations learned by a vision encoder align with those of a language model, providing deeper insights into cross-modal learning and feature alignment.

## C SIMILARITY PRESERVING LOSS FUNCTION

Our alignment of vision and language representations follows a distinct approach. Unlike contrastive losses commonly used in VLMs that rely on large datasets for effective convergence, we adopt a knowledge distillation-inspired method using a similarity-preserving loss (Tung & Mori, 2019) to guide the image encoder with insights from the language model. Originally developed for a student-teacher framework, this loss builds on the principle that semantically similar inputs elicit similar

Table 7: Hyperparameters for continual learning analyses: All tasks are trained for 50 epochs with SGD optimizer for ExLG method and AdamW for ImLG.

| Method | Seq-CIFAR10 | Seq-TinyImageNet | DN4IL |
|---|---|---|---|
| ExLG | $lr = 0.05$ $\lambda = 50.0$ | $lr = 0.05$ $\lambda = 100.0$ | $lr = 0.03$ $\lambda = 100.0$ |
| ImLG | $lr = 0.001$ | $lr = 0.0001$ | $lr = 0.0001$ |

activation patterns in trained neural networks. In a knowledge distillation setting, the goal is for the trained teacher network to provide additional supervision to train a student network effectively.

In our framework, this loss (in Equation 2) ensures that inputs with similar semantic meanings in the language model induce correspondingly similar activations in the vision encoder, thereby fostering a shared representation space. By leveraging pre-trained language embeddings as a reference, the similarity-preserving loss enables the vision encoder to learn high-level, semantically rich features that transcend superficial correlations. Specifically, this loss supervises the vision encoder by comparing pairwise activation similarities within each batch and penalizing discrepancies in their similarity matrices. This approach bridges the textual and visual domains, enabling robust cross-modal learning with minimal additional training complexity.

## D  RELATED WORKS

### Multi-modal Learning

Vision-Language Models (VLMs) focus on learning joint vision-and-language representations for tasks like visual question answering, visual reasoning, captioning and retrieval. CLIP (Radford et al., 2021) aligns vision and language embeddings through contrastive learning on large-scale multimodal data. BLIP (Li et al., 2022) fuses vision and language data during training, effectively integrating modalities to perform multi-modal tasks such as captioning and visual reasoning. LLaVA (Liu et al., 2024) expands these capabilities by instruction tuning large models to create multimodal chat assistants. Many vision-language models, such as CLIP, rely on contrastive losses to align embeddings by training dual encoders for images and text. These encoders are trained on large multimodal datasets, matching vector representations across large batches to compute similarity effectively. Classification tasks in such models are formulated as retrieval problems, where during inference, the class name with the closest match in the embedding space is retrieved. These models also often face challenges in generalizing to images outside their pre-training datasets, requiring additional fine-tuning techniques or adaptations to handle diverse data distributions effectively.

Our approach diverges from these paradigms by focusing on a setup of vision encoder, classifier doing supervised learning with cross-entropy loss, without contrastive loss or retrieval-based prediction. We focus on learning visual representations from scratch for visual tasks by leveraging pre-trained language models as guidance in different ways, eliminating the need for large-scale multi-modal datasets or computationally expensive joint training. We venture beyond the current paradigms of joint vision-and-language pre-training or parameter-efficient fine-tuning (PEFT). Instead, our work uniquely uses language guidance as a modular component to enhance visual learning, evaluated across fundamental tasks requiring robustness, generalization, and adaptability.

**Language Guidance** Recent studies have explored leveraging language-vision alignments to improve representation learning. One line of research investigates using encoded image captions as semantic signals to enhance contrastive learning. For instance,(El Banani et al., 2023) propose a sampling method that identifies linguistically similar image pairs using caption embeddings. So in this method demonstrates they leverage language to identify similar images in the batch over traditional augmentation-based approaches. In Sariyildiz et al. (2020) the goal is to have many different proxy tasks conditioned on vision and language that such that solving these tasks will help learn better representations. The first involves predicting image tags from captions, and the second employs the image-conditioned masked language modeling task. The framework involves multiple passes, requires high-quality, paired image-caption datasets additional annotations for proxy tasks. (Stroud et al., 2020) encodes video metadata using a BERT-based text encoder and trains a video model

Table 8: Results with a bigger CNN backbone- ResNet50

|  | CIFAR10 | | | CelebA | | |
|---|---|---|---|---|---|---|
|  | Base | ExLG | ImLG | Base | ExLG | ImLG |
| ResNet50 | 94.38 | **97.21** | **95.86** | 63.88 | **74.45** | **76.58** |

to predict these embeddings. The approach assumes that metadata (e.g., titles, descriptions) provides weak supervision for learning semantic video representations, and this approach is primarily designed for pre-training video models.

Further, (Merullo et al., 2022) investigate whether simple linear transformations can map image features to text space effectively. Their method involves training a linear projection from vision encoder outputs to a shared text embedding space, achieving notable results in cross-domain retrieval tasks. Other works (Tsimpoukelli et al., 2021) use image-text pairs to pre-train vision encoders through a captioning task, freezing the language encoder and using gradients only to update the vision encoder. Despite its innovative approach, Frozen's performance is limited, as noted in the paper, and does not achieve higher results consistently across tasks. There are studies that try to establish pre-training based on image-captioning task. However, all often neglect a deeper analysis of the vision backbone's properties in isolation.

A few works provided insights that also guided our design. The Platonic Representation Hypothesis (Huh et al., 2024) posits that representations learned by neural networks across different modalities, objectives, or architectures tend to converge toward shared high-level abstractions. Studies such as (Maniparambil et al., 2024) investigate the extent to which vision and language models encode similar concepts, which is critical for cross-modal learning and alignment. Research by (Schwettmann et al., 2023) further shows that text transformers, even when trained exclusively on text data, develop multi-modal neurons—neurons that respond similarly to semantically related image and text embeddings. Additionally, (Pang et al., 2023) highlight the versatility of incorporating vision encodings into a language encoder. They posit that the language blocks act as a filter, amplifying relevant features and distilling essential information from visual inputs, showcasing their potential for enhancing cross-modal learning.

Our aim was to investigate the properties of vision encoders under the influence of language supervision. Specifically, we sought to understand when and how language guidance impacts the learning of image representations and to explore distinct strategies for integrating semantic information through Explicit Language Guidance (ExLG) and Implicit Language Guidance (ImLG). We enable the vision encoders to learn better semantic concepts and produce more robust representations and we test it on several key challenges, including shortcut learning, adversarial robustness, texture bias, out-of-distribution (OOD) generalization, and continual learning. Unlike prior works, which do not focus on these vision-based challenges, our framework aims to offer a different perspective on how language can guide visual representations to tackle these challenges.

# E  ADDITIONAL RESULTS

## E.1  DIFFERENT IMAGE ENCODERS

In this section, we present additional results, starting with an evaluation of different vision encoders across various datasets. Table 8 provides results for both the ExLG and ImLG methods using the ResNet50 vision encoder. Additionally, we evaluate performance on a larger dataset (ImageNet100) using both CNN- and transformer-based vision encoder architectures, as shown in Table 9. As the dataset complexity and vision model size scale up, we observe more significant improvements, demonstrating the scalability of our methods.

## E.2  LANGUAGE DESCRIPTIONS

We conduct ablation studies with different types of descriptions used in ExLG. Note that the descriptions are class-specific, not image-specific, thus we need descriptions as the number of classes

Table 9: Results with a bigger dataset- ImageNet100

|  | ImageNet100 | | |
|---|---|---|---|
|  | Base | ExLG | ImLG |
| ResNet18 | 71.46 | **79.42** | **72.37** |
| VIT | 54.77 | **56.16** | **55.06** |

Table 10: Samples of few descriptions of classes of CIFAR10 dataset.

| Class | Language Descriptions | | |
|---|---|---|---|
|  | Used in ExLG | Simple | Random |
| "bird": | "A small, feathered vertebrate with two wings for flight, a beak, and typically two legs" | "This is a bird" | "The photo of an object or entity" |
| "cat" | "A small, agile mammal with a slender body, sharp claws, whiskers, and a long tail" | "This is a cat" | "The photo of an object or entity" |
| "cargo ship" | "A large vessel/boat with a flat deck, towering cranes, and stacked containers in sea and harbor." | "This is a cargo ship" | "The photo of an object or entity" |

and not images. Table 10 compares various types of descriptions, with detailed results in Table 11.Our findings show that detailed descriptions with rich semantic context lead to the highest gains. Simpler descriptions also provide improvements over the baseline, albeit to a lesser extent, while random descriptions offer minimal benefit.

### E.3 DIFFERENT LANGUAGE MODELS

In this section, we evaluate the impact of different language models (LMs) on our framework. In the experiments presented in the main paper, we utilize the LM - Sentence Transformer (all-MiniLM-L6-v2) (Reimers & Gurevych, 2019), an efficient model with only 22.7M parameters, which adds minimal computational overhead. To further investigate, we conduct experiments using a (1) Larger LM - "all-distilroberta-v1" with 82.1M parameters. Additionally, we examine two alternative setups: (2) LM-Rand - a Sentence Transformer model with random weight permutation (all-MiniLM-L6-v2) (3) LM-Code - a CodeBERT (Feng et al., 2020) model trained on programming languages (CodeLM)

While larger models yield improved performance, efficient models like all-MiniLM-L6-v2 are sufficient, provided that the descriptions are semantically rich. Models with random weights or trained on unrelated domains (e.g., CodeLM) act as mild regularizers but perform significantly worse than semantically trained language models. The only scenario where they surpass the baseline is in

Table 11: Results with different language descriptions using ExLG method

|  |  | Classification | Continual Learning |
|---|---|---|---|
|  |  | CIFAR10 | DN4IL |
|  | Base | 94.84 | 24.15 |
|  | ExLG | **95.12** | **27.71** |
| Language Descriptions | Simple desc | 94.92 | 24.74 |
|  | Random Desc | 92.47 | 18.86 |

Table 12: Results with different language models using ExLG method.

| | | Cls | | Shortcut | OOD | | | CL |
|---|---|---|---|---|---|---|---|---|
| | | CIFAR10 | TinyImg | CelebA | ImgNet-O | ImgNet-R | ImgNet-A | DN4IL |
| | Base | 94.84 | 58.73 | 61.28 | 41.73 | 10.59 | 1.92 | 24.15 |
| ExLG | LM | **95.12** | 65.63 | 72.11 | 46.70 | **14.95** | **2.94** | **27.71** |
| | Larger LM | 95.01 | **65.89** | **72.93** | **47.51** | 14.65 | 2.92 | 26.84 |
| | LM-Rand | 92.17 | 58.02 | 67.24 | 40.65 | 9.62 | 2.03 | 22.08 |
| | LM-Code | 93.69 | 58.93 | 67.14 | 38.50 | 9.40 | 2.50 | 21.76 |

Table 13: Results with different language models using ImLG method.

| | | Cls | | Shortcut | OOD | | | CL |
|---|---|---|---|---|---|---|---|---|
| | | CIFAR10 | TinyImg | CelebA | ImgNet-O | ImgNet-R | ImgNet-A | DN4IL |
| | Base | 94.84 | 58.73 | 61.28 | 41.73 | 10.59 | 1.92 | 24.15 |
| ImLG | LM | 93.41 | 60.02 | 75.90 | **42.20** | 12.10 | 2.37 | 24.22 |
| | Larger LM | **93.78** | **61.16** | **77.55** | 42.12 | **12.71** | **2.45** | **24.51** |
| | LM-Rand | 92.50 | 57.98 | 67.41 | 28.45 | 6.55 | 1.50 | 19.85 |
| | LM-Code | 92.00 | 47.83 | 68.82 | 30.62 | 6.67 | 1.89 | 20.54 |

the CelebA dataset (shortcut learning). In the case of CelebA, which is highly sensitive with only two classes, random weights or CodeLM provide some regularization benefits. However, their performance does not match that of language models trained on natural language, underscoring the importance of semantic context.

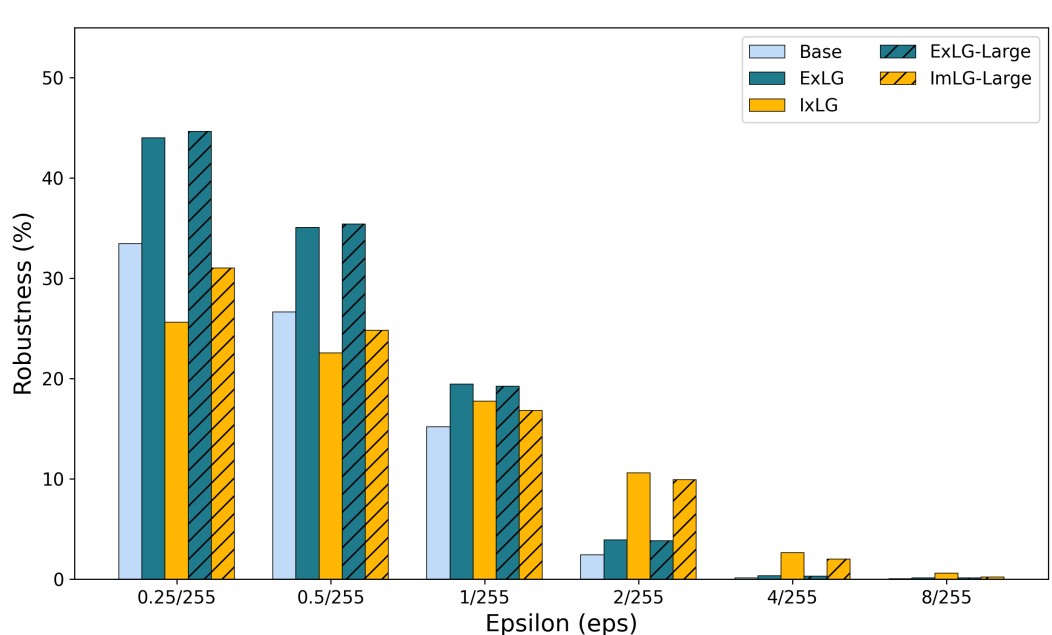

Figure 9: Adversarial Robustness - PGD10 attack on CIFAR10 using larger language model

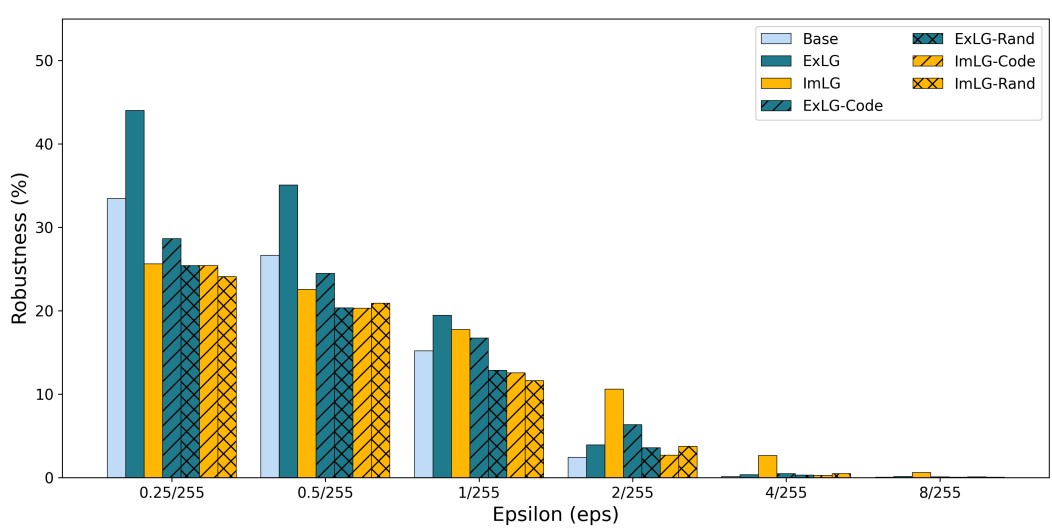

Figure 10: Adversarial Robustness - PGD10 attack on CIFAR10 using random and codeLM language model

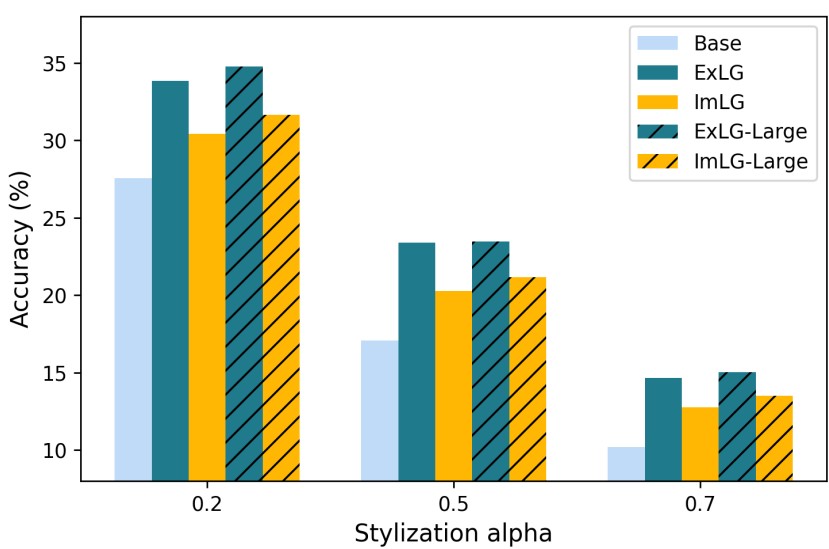

Figure 11: Texture Bias - results on Stylized TinyImageNet using larger language model

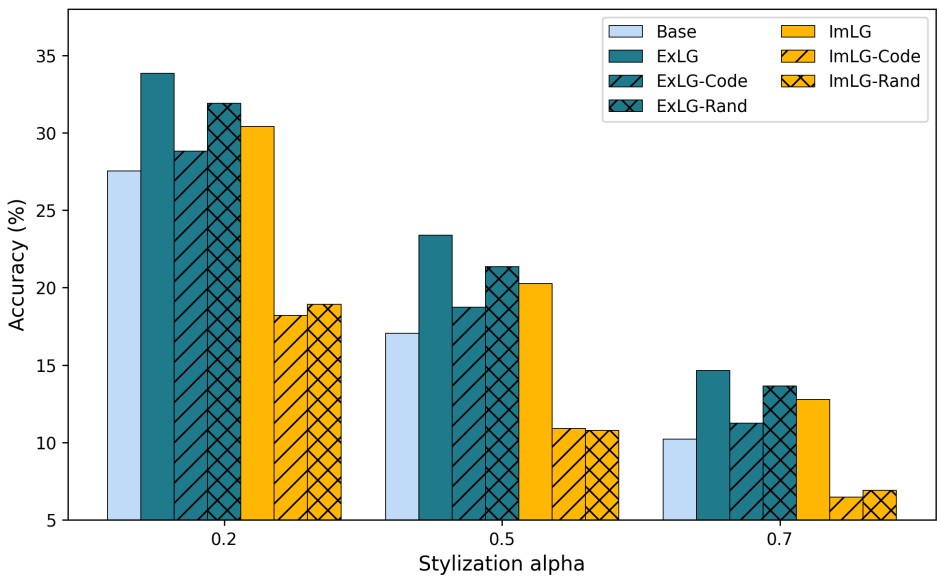

Figure 12: Texture Bias - results on Stylized TinyImageNet using random and CodeLM language model

