# OpenReview forum: "Language Guided Representation Learning"
_ICLR.cc/2025/Conference — Submitted to ICLR 2025_

### Official Review · Reviewer_XVTH · 2024-10-29

**Soundness:** 2
**Presentation:** 3
**Contribution:** 2
**Rating:** 5
**Confidence:** 4

**Summary:**

The paper asks whether non-visual language models provide advantages when used to create an image representation, over a traditional end-to-end image classifier alone. It compares a traditional classifier with a proposed “ExLG” vision encoder and an “ImLG” encoder, where ExLG adds a tung-mori student-teacher setup to align the image representation with a language representation, and ImLG incorporates a frozen pretrained language model as the final stage of the vision encoder.  With this setup, they train on several standard classification problems and compare performance on low-data regimes, OOD generalization, strongly biased data, and adversarial robustness, and continual learning. They conclude that the incorporation of pretrained language models is helpful in all these settings.

**Strengths:**

The community has a lot of interest in the language models’ capabilities to represent the visual world without ever having been exposed to an image during training. The paper poses natural questions, looking beyond basic classification performance to ask whether incorporating a language model improves the inductive biases for a model. The benchmark datasets used to investigate OOD and bias behavior are reasonable, and the robustness test is appropriate.

**Weaknesses:**

The paper as currently presented doesn’t have enough evidence to support its broad claims. The claim is that several types of robustness improve when language modeling is incorporated (implying that there are benefits from having the “knowledge” derived from lots of text training), but there are several possible confounders that aren’t investigated.

For example, the addition of extra loss terms (for ExLG) or extra layers (for ImLG) could have a regularizing effect regardless of the specific content of those extras, which could mean that there is nothing special about language knowledge.  The paper would be strengthened if it presented clearer evidence that the essential benefits come from the fact that the extra model involved is a language model trained on lots of text, as compared to e.g., a random neural network. Ablations are needed on: what type of text is used in ExG; how powerful the language model is; whether it matters if the language model is trained on natural text or some non-object related task, or left uninitialized. Ideally the hyperparameters can be held fixed while comparing the use of a language model to a baseline with the same computational form that doesn’t have the benefit of large-scale text training.

The paper’s investigations are related to the idea articulated in the recent paper “The Platonic representation hypothesis” by Huh, and it would be nice to cite+connect it.

**Questions:**

In the ExG case, it is unclear what text is used for aligning the representation. Is it image-specific text, or class-generic text, or something else; how was it chosen, and how important is this choice? What would be the effect of some text-per-class that has nothing to do with the class?

As the language model becomes more powerful, does it improve OOD, resistance to bias, robustness, and CI behavior? Only one small model size comparison is done, and only on basic classification accuracy.

The claim is that incorporating a language model helps, but does it need to be a language model pretrained on real text?  Would performance benefits be obtained from a randomly-initialized language model? What about an early checkpoint of a language model with poor performance, or a model trained on non-visual-world-descriptive text such as a code LM?

---

> ### Author Response · Authors · 2024-11-22
> **Response to Reviewer XVTH**
>
> We would like to thank the reviewer for finding our work useful and providing thoughtful feedback. Below, we address your questions and provide detailed responses.
> * We agree on the importance of ruling out potential confounders to isolate the effects of language guidance. We have incorporated the following ablations in line with your valuable suggestions.:
>     * Text Descriptions: For ExLG, we experimented with simple descriptions which have the class description in the template “This is a <class_name>” and random descriptions like, “this is an image of an entity”.
>     * Language Model weights - We also perform 2 experiments, one with language model initialized with random weights, and another by loading a codeLM model (CodeBERT trained on programming language). The results are below (and also in the Appendix of the paper) for CIFAR10-classification and DN4IL-continual learning for both (a) and (b).
> Results show that semantically rich descriptions and language models trained on text data outperform other descriptions and models in both classification and continual learning tasks.
>     * Language Model Size and Power: We also are experimenting with language model of bigger size to observe if more powerful language models contribute greater benefits, and will be updated shortly. We meanwhile also ran experiments with bigger vision encoders (ResNet50, VIT) and bigger data (ImageNet), both are tabulated in Appendix.
>
> |                            |                           | Classification | Continual Learning |
> |----------------------------|---------------------------|----------------|--------------------|
> |                            |                           | CIFAR10        | DN4IL              |
> |                            | Base                     | 94.84          | 24.15              |
> |                            | ExLG                     | **95.12**      | **27.71**          |
> | **Language Descriptions**  | Simple desc              | 94.92          | 24.74              |
> |                            | Random Desc              | 92.47          | 18.86              |
> | **Language Model Weights** | Random Weight            | 92.17          | 22.08              |
> |                            | CodeLM                   | 93.69          | 21.76              |
>
>  - We thank the reviewer for highlighting the insightful paper. The Platonic Representation Hypothesis posits that different models converge to similar representations of high-level abstractions across diverse modalities and tasks. We have incorporated this theoretical connection in the revised version.

---

> > ### Comment · Reviewer_XVTH · 2024-11-25
> > **Clarify the answer to a couple questions**
> >
> > I appreciate the added information on ablations.
> >
> > I may have missed this detail in the explanation but did not see it: Can you clarify for my understanding, which text is used for each image in the full ExG case?
> > (1) Is it image-specific text (for example, is it text that differs between images in the same class?)
> > (2) or class-generic text (i.e., text that depends only on the class and not the image)
> > (3) or something else
> > And how was that text chosen?
> > For example, in figure 2a, I cannot tell whether the example text is text you actually use [or just illustrative], or if it is (2) or (1).  The text differs slightly from the text in figure 1; is this a typo, or is the variation created by some aspect of the method?
> >
> > On the new ablations. You have two methods ExLG and ImLG, and you seem to be ablating just one.  Which one?  You should label it in the table, e.g., "ExLG with CodeLM" or "ImLG with CodeLM".  Because you are making claims about both approaches, you should explore those same ablations of both of your methods instead of just one method.
> >
> > Also, the ablations in the new table show the effects on continual learning but do not explore the effects on your other claimed benefits such as OOD, robustness, shortcuts, etc.  To defend the claims and show that each of those benefits come from your methods, you should include these measurements.
> >
> > Is the architecture identical for Sentence-BERT and CodeLM and random weights?
> >
> > The citation for Reimers 2019 - you should cite the peer-reviewed venue EMNLP 2019 because it's not just a preprint.

---

> > > ### Author Response · Authors · 2024-11-25
> > >
> > > Thank you for your prompt response.
> > >
> > > - Regarding the test, the text used in experiments is (2) class-generic text that depends only on the class and not on the image. The text in Figures 1 and 2, was more illustrative (manually written). A GPT model was used to generate descriptions and some sample examples are shown in Table 11. The text was generated and kept the same for all experiments.
> > > - Ablation - The table right now only shows for ExLG. I apologize for that, we are also doing the same for ImLG and will update the table accordingly. About the datasets, we chose one data in classification and one in Continual learning for ablatioin. However, we will provide all the other anlaysis as well.
> > > - Arch - For Sentence transformer, we had chosen the efficient version - MiniLM-L6-v2 (22M parameters)
> > > For random weights, we use the same model but reinitialize all the weights. For CodeLM, we chose a SentenceTransformer - CodeBERT (100M).
> > >
> > > Will upload all the results and changes together and update you here shortly.

---

> > > > ### Comment · Reviewer_XVTH · 2024-11-26
> > > > **If posting a new version**
> > > >
> > > > If/when posting a new version, help us out by pointing out here which sections to look at for the updates.  Thank you!

---

> > > > > ### Author Response · Authors · 2024-11-27
> > > > >
> > > > > I have updated the Appendix of the paper with all the results - Section E.2 and E.3 - Tables 11, 12,13 and Figures 9-12 for all the analysis of ablations. All the updates are in blue. Thank you.

---

> > > > > > ### Author Response · Authors · 2024-12-02
> > > > > >
> > > > > > We again thank the reviewer for their feedback to help improve the quality of our work.
> > > > > > \
> > > > > > As a summary, our work focuses on supervised learning with image-label pairs - a discriminative classifier, exploring the under-researched area of leveraging language as structured knowledge to address challenges such as texture bias, shortcut learning, OOD generalization, robustness, and catastrophic forgetting. Through comprehensive ablations and analyses (detailed in the Appendix), we show the impact of different language guidances. By focusing on core supervised learning challenges often overlooked in scaling-focused literature and conducting extensive analyses, we aim to provide a holistic perspective and valuable insights. We have summarized the motivation and provided additional details about the new results in the general responses, and we kindly invite the reviewer to see if it helps address their concerns and increase support for our paper.

---

### Official Review · Reviewer_Q65P · 2024-10-29

**Soundness:** 2
**Presentation:** 2
**Contribution:** 2
**Rating:** 5
**Confidence:** 3

**Summary:**

This paper studied the effect of additional language information in image tasks training. The research found out that language guidance in the form of explicit representation alignment and implicit access to the language model improved ResNet's
- OOD generalization
- shortcut learning (reduction of spurious correlations)
- bias on textures
- robustness against adversarial attacks
- continual learning (reduction of catastrophic forgetting)

**Strengths:**

1. Thoroughly studied the role of representation learning in language models.
2. Identified that the guidance of language reduced many unwanted behaviors in image models training, such as catastrophic forgetting and the vulnerability against adversarial attacks.

**Weaknesses:**

1. The evaluation needs to be fair. For all the experiments, we should keep the total (frozen) parameter count the same, and (ideally) the tunable parameter count the same. Otherwise we may attribute the better performance of ExLG/ImLG comparing to the baseline models to the increase of the number of parameters, not language-guidance.
2. You should add more baselines/ablations. For example, in the "continual learning" experiment, it is unclear whether the reduction of catastrophic forgetting from ER to ExLG/ImLG is due to the language guidance or access to a large bank of information (be it language/image/...).
3. You should conduct a more thorough related work analysis. I haven't conducted a thorough literature review on the topic, but this paper should have a "related work" section that distinguishes it from other related concepts or frameworks, such as CLIP.
4. There is a lack of explanation or insight into how the language guidance improved the image's performance on these tasks. Consider how the alignment loss improved the representation alignment by offering some interpretability analysis, such as probing the learned resnet's inner representations, https://arxiv.org/pdf/2410.06940 applies on diffusion models, but did several layer-by-layer analyses, which I think is valuable at improving the insights of your work.

**Questions:**

1. Could you clarify again what is the core difference between your approach with other language/vision representation alignment approaches, such as CLIP? Especially for the explicit guidance.
2. In section 5.3 and figure 4, what is the "stylization alpha"? It seems that this is not explained.
3. In figure 7, how did you calculate the "plasticity" and "stability"? How are these precisely defined mathematically?

---

> ### Author Response · Authors · 2024-11-21
> **Response to reviewer Reviewer Q65P**
>
> We would like to thank the reviewer for providing valuable feedback. Below, we provide our responses to the questions.
>
> - Ablation - We appreciate the emphasis on ensuring fairness. To address some of the confounding factors, we conducted additional ablation studies to isolate the effects of language guidance. We experimented with various descriptions, including: Simple templates: “This is a <class_name>.” Random phrases: “This is an image of an entity.” We also tested different language models - language models initialized with random weights and - Pre-trained CodeLM models (e.g., CodeBERT trained on programming language corpora). Results (Table 11) indicate that detailed, semantically meaningful descriptions and language models pre-trained on text lead to better performance.
>
> For ExLG, the additional component is the frozen language model to get the description for the alignment loss, and tunable parameters is the vision encoder (baseline) and introduces no additional trainable parameters. For ImLG, the language model serves as implicit guidance by adding it post the vision encoder block. A linear layer projects the vision features to the dimensions required by the language block.. As removing these components would invalidate the approach itself, we welcome further suggestions on alternative ablations that could strengthen this analysis. Also in continual learning, we want to clarify that no additional information bank or external data is used compared to baseline.
>
> - Related works - We have added a detailed Related Work section in the appendix, which distinguishes our approach from prior works, including: Multi-modal training frameworks that focus on joint vision-language embeddings and also few works on Vision and Language alignment. Kindly also view the Generic response above, If you have further suggestions for additional ablations, we would be happy to incorporate them.
>
> - The "stylization alpha" controls the strength of style transfer applied to images. It determines the extent to which the original image's texture is replaced with the style features from a reference image, while retaining the underlying object shape. This method is based on Stylized-ImageNet [1], where increasing the alpha value results in progressively higher texture modifications. In our experiments, we use different levels of alpha to test the models' ability to generalize when texture bias is mitigated
>
> - Plasticity: Measures the model's capability to learn new tasks. It is calculated as the average accuracy of each task when it is first learned (e.g., the accuracy of the network trained on task
> 𝑇2 , evaluated on the test set of 𝑇2
> Stability: Measures the model's ability to retain knowledge from previously learned tasks. It is computed as the average accuracy of all tasks 1 to  𝑇 − 1 after learning the final task  𝑇
> Trade-off: To assess the balance between plasticity and stability, we use the following metric:
> The Trade-off metric  is defined as: 2 x P x S /((P + S)
>
> where:
> - Plasticity (P): The average accuracy of each task when it is first learned.
> - Stability (S) :The average accuracy of all tasks \(1 : T-1\) after learning the final task \(T\).
> All of these are updated in the paper as well.
>
> - Thank you for the suggestion. For interpretability, we performed GradCAM analysis.It computes gradients of the output with respect to specific feature maps, indicating which regions of the image the model focuses on for decision-making. This is especially useful for understanding how different layers in a model process and prioritize visual features. In Figure 3, activation maps demonstrate how language-guided models reduce reliance on spurious correlations, such as background or superficial cues, compared to baseline models, and instead focus on task-relevant features.
> For ImLG, as we add an Language block on top of a vision encoder, activation maps after each block are provided in Figure 8, showing progressive refinement of task-relevant features. Did you mean for us to perform linear probing specifically on the vision encoder layers for both ExLG and ImLG models? If so, could you clarify whether the goal is to assess the impact of alignment losses on intermediate representations? We’d like to ensure we address your suggestion appropriately.

---

> > ### Author Response · Authors · 2024-11-30
> >
> > To further address the reviewer’s request for insights into how language guidance improves performance -  Our primary objective was to induce semantic knowledge into the vision encoder, enabling it to learn richer representations. To demonstrate this, we initially presented GradCAM visualizations on trained models to show the focused on relevant regions of the image (Figure 3 and 8). Below, we perform a similarity analysis of feature embeddings for images of the same class but across different domains. Same as Figure 1, we selected images from the same class but across different domains, requiring the model to learn shared features while ignoring domain-specific background. However, as observed in domain-incremental learning experiments within the continual learning setup, performance deteriorates across tasks.
> >
> > To delve deeper, we conducted a similarity analysis of feature embeddings for these images using our trained models in multiple layers. (For reference the plot is similar to the one on Figure1, as we cant show the plots, we add the matrix below).
> > Our findings demonstrate that language guidance significantly enhances cross-domain alignment, in all layers of ResNet18 network, thus further substantiating the supervision from language to produce more rich representations.
> >
> > ResNet18 - block 3
> > \
> > BASE
> > \
> > | Domain | real | clipart | infograph | painting | sketch | quickdraw |
> > \
> > | real | 1.00 | 0.00 | 0.00 | 0.00 | 0.00 | 0.00 |
> > \
> > | clipart | 0.45 | 1.00 | 0.00 | 0.00 | 0.00 | 0.00 |
> > \
> > | infograph | 0.23 | 0.51 | 1.00 | 0.00 | 0.00 | 0.00 |
> > \
> > | painting | 0.31 | 0.40 | 0.19 | 1.00 | 0.00 | 0.00 |
> > \
> > | sketch | 0.19 | 0.55 | 0.46 | 0.16 | 1.00 | 0.00 |
> > \
> > | quickdraw | 0.25 | 0.73 | 0.51 | 0.14 | 0.77 | 1.00 |
> >
> > ExLG
> > \
> > | Domain | real | clipart | infograph | painting | sketch | quickdraw |
> > \
> > | real | 1.00 | 0.00 | 0.00 | 0.00 | 0.00 | 0.00 |
> > \
> > | clipart | 0.46 | 1.00 | 0.00 | 0.00 | 0.00 | 0.00 |
> > \
> > | infograph | 0.35 | 0.54 | 1.00 | 0.00 | 0.00 | 0.00 |
> > \
> > | painting | 0.35 | 0.64 | 0.27 | 1.00 | 0.00 | 0.00 |
> > \
> > | sketch | 0.26 | 0.54 | 0.42 | 0.31 | 1.00 | 0.00 |
> > \
> > | quickdraw | 0.32 | 0.72 | 0.54 | 0.32 | 0.79 | 1.00 |
> > \
> >
> > ImLG
> > \
> > | Domain | real | clipart | infograph | painting | sketch | quickdraw |
> > \
> > | real | 1.00 | 0.00 | 0.00 | 0.00 | 0.00 | 0.00 |
> > \
> > | clipart | 0.43 | 1.00 | 0.00 | 0.00 | 0.00 | 0.00 |
> > \
> > | infograph | 0.41 | 0.51 | 1.00 | 0.00 | 0.00 | 0.00 |
> > \
> > | painting | 0.56 | 0.69 | 0.38 | 1.00 | 0.00 | 0.00 |
> > \
> > | sketch | 0.20 | 0.65 | 0.48 | 0.40 | 1.00 | 0.00 |
> > \
> > | quickdraw | 0.26 | 0.79 | 0.46 | 0.46 | 0.86 | 1.00 |
> > \
> >
> > ResNet8 -block1
> > \
> > Base
> > \
> > | Domain     | real | clipart | infograph | painting | sketch | quickdraw |
> > \
> > | real | 1.00 | 0.00 | 0.00 | 0.00 | 0.00 | 0.00 |
> > \
> > | clipart | 0.03 | 1.00 | 0.00 | 0.00 | 0.00 | 0.00 |
> > \
> > | infograph | 0.02 | 0.13 | 1.00 | 0.00 | 0.00 | 0.00 |
> > \
> > | painting | 0.07 | 0.37 | 0.03 | 1.00 | 0.00 | 0.00 |
> > \
> > | sketch | 0.07 | 0.12 | 0.36 | 0.05 | 1.00 | 0.00 |
> > \
> > | quickdraw | 0.20 | 0.21 | 0.28 | 0.01 | 0.85 | 1.00 |
> > \
> >
> > ExLG
> > \
> > | Domain     | real | clipart | infograph | painting | sketch | quickdraw |
> > \
> > | real | 1.00 | 0.00 | 0.00 | 0.00 | 0.00 | 0.00 |
> > \
> > | clipart | 0.04 | 1.00 | 0.00 | 0.00 | 0.00 | 0.00 |
> > \
> > | infograph | 0.04 | 0.20 | 1.00 | 0.00 | 0.00 | 0.00 |
> > \
> > | painting | 0.08 | 0.42 | 0.04 | 1.00 | 0.00 | 0.00 |
> > \
> > | sketch | 0.05 | 0.31 | 0.66 | 0.04 | 1.00 | 0.00 |
> > \
> > | quickdraw | 0.11 | 0.32 | 0.71 | 0.02 | 0.93 | 1.00 |
> > \
> >
> > ImLG
> > \
> > | Domain     | real | clipart | infograph | painting | sketch | quickdraw |
> > \
> > | real | 1.00 | 0.00 | 0.00 | 0.00 | 0.00 | 0.00 |
> > \
> > | clipart | 0.01 | 1.00 | 0.00 | 0.00 | 0.00 | 0.00 |
> > \
> > | infograph | 0.03 | 0.16 | 1.00 | 0.00 | 0.00 | 0.00 |
> > \
> > | painting | 0.07 | 0.32 | 0.02 | 1.00 | 0.00 | 0.00 |
> > \
> > | sketch | 0.02 | 0.33 | 0.40 | 0.06 | 1.00 | 0.00 |
> > \
> > | quickdraw | 0.02 | 0.53 | 0.45 | 0.01 | 0.87 | 1.00 |
> > \

---

> > > ### Author Response · Authors · 2024-12-02
> > >
> > > We again thank the reviewer for their feedback to help improve the quality of our work.
> > > \
> > > As a summary, our work focuses on supervised learning with image-label pairs - a discriminative classifier, exploring the under-researched area of leveraging language as structured knowledge to address challenges such as texture bias, shortcut learning, OOD generalization, robustness, and catastrophic forgetting. Through comprehensive ablations and analyses (detailed in the Appendix), we show the impact of different language guidances. By focusing on core supervised learning challenges often overlooked in scaling-focused literature and conducting extensive analyses, we aim to provide a holistic perspective and valuable insights. We have summarized the motivation and provided additional details about the new results in the general responses, and we kindly invite the reviewer to see if it helps address their concerns and increase support for our paper.

---

### Official Review · Reviewer_QcmS · 2024-10-30

**Soundness:** 2
**Presentation:** 3
**Contribution:** 1
**Rating:** 3
**Confidence:** 4

**Summary:**

The authors investigate using natural language to enhance visual representations, and how this enhancement affects systematic generalization and catastrophic forgetting in neural networks. More specifically, inspired by human cognition, the authors propose that language, as a tool for abstraction and concept-sharing, can help guide DNNs to better, more abstract representation learning. The authors explore two main approaches: Explicit Language Guidance (ExLG), which aligns visual representations with high-level language descriptions, and Implicit Language Guidance (ImLG), where a pre-trained language model “indirectly” enhances the vision model. Both methods are tested extensively across diverse tasks such as generalization to new data (IID and OOD), among others. Perhaps unsurprisingly, the results show improvements over baseline models. ExLG performed better on generalization tasks, while ImLG showed advantages in robustness and shortcut learning. As seminal work in the past (e.g., CLIP), the study highlights language guidance as a powerful tool for creating models that generalize and retain knowledge more effectively.

**Strengths:**

1. The paper is well written and clearly explained
2. Figures are clear and informative.
3. The topic is timely and of extensive interest and applicability.

**Weaknesses:**

The main, and crucial weakness of this work is its novelty and scope. Although, as the authors point out, their method slightly differs from other VLMs whose representation “fuse” vision and language embeddings, both the added theoretical and empirical value of this paper is poor:

1. Other papers already make the point that language can generate richer representations that have an impact on the issues highlighted by the authors.

2. I would compare the proposed methods with other VLM models, in order to show concrete empirical value of this paper.

**Questions:**

*If you mention the Global Workspace Theory, I think it’s only fair to cite at the very least Dehaene (1998) and I would also include Baars (1994).
*Line 123, CKA is first presented without spelling the acronym.
*I would define what a “conventional classification model” is.

Typo:
Line 036: “…is one of the aspects of human cognition is still a challenge for neural networks...” -> that is still
Line 049: “...context of continual learning (?)...”  issue with citation.
Line 099: “...System 2 (Explicit) processing (Daniel, 2017).” Citation should be Kahneman.

---

> ### Author Response · Authors · 2024-11-21
> **Response to Reviewer QcmS**
>
> We appreciate the reviewer's observations and feedback.
>
> - Novelty -
> While we acknowledge that many prior works with vision-language models, they rarely address fundamental challenges faced by vision encoders, such as shortcut learning, texture bias, adversarial robustness, and catastrophic forgetting. We  avoided direct comparisons with existing VLMs such as CLIP or BLIP, which are optimized for large-scale multimodal tasks and operate with computational and data requirements and also predominantly focus on tasks like retrieval, and visual question answering.  Instead, our work is rooted in evaluating how language, as a source of structured semantic knowledge, can enhance vision encoders across key challenges. Unlike prior studies, we focus on understanding these effects in a basic classification framework , comprising a vision encoder network paired with a classifier, trained on an image dataset using supervised learning with a cross-entropy loss, without introducing multi-modal datasets or retrieval-based tasks.
>
> Kindly refer to the Generic Response and the expanded Related Works section in the revised version for further clarification.
>
> - Thank you for the relevant papers. The other comments have been addressed and incorporated into the revised version of the paper.

---

> ### Comment · Reviewer_QcmS · 2024-12-02
>
> Thank you for the response. I raise my score to 4 (but below 5). I still believe that the lack of novelty warrants a publication in a more specialized workshop, and not a main contribution in ICLR.

---

> > ### Author Response · Authors · 2024-12-02
> >
> > We again thank the reviewer for their feedback to help improve the quality of our work.
> > \
> > As a summary, our work focuses on supervised learning with image-label pairs - a discriminative classifier, exploring the under-researched area of leveraging language as structured knowledge to address challenges such as texture bias, shortcut learning, OOD generalization, robustness, and catastrophic forgetting. Through comprehensive ablations and analyses (detailed in the Appendix), we show the impact of different language guidances. By focusing on core supervised learning challenges often overlooked in scaling-focused literature and conducting extensive analyses, we aim to provide a holistic perspective and valuable insights. We have summarized the motivation and provided additional details about the new results in the general responses.
> >
> > We are a bit unclear, as there does not appear to be a score of 4 in the rating scale. We would request the reviewer to share any additional feedback or suggestions on how we can better address your concerns and potentially increase your support for our work.

---

### Official Review · Reviewer_W4mR · 2024-11-03

**Soundness:** 2
**Presentation:** 2
**Contribution:** 2
**Rating:** 5
**Confidence:** 4

**Summary:**

The paper studies language guided representation learning and its potential techniques for incorporating language guidance into vision representation learning. The paper considers two techniques for incorporating language guidance, one based on explicit guidance (ExLG) and the other based on implicit guidance (ImLG). The paper investigates the effect of language guidance on sample efficiency, OOD generalization, spurious feature learning, shortcut learning, and robustness. Generally, the paper finds ExLG improves on all aspects over traditional approaches for performing vision representation learning.

**Strengths:**

A thorough improvement from language guidance: I found the paper to do a reasonably good exploration of language guidance with strong results.

Consistent and large amount of experimentation: The paper has many experiments comparing its two proposed methods.

Interesting analysis: Figures 3, 4, and 5 show some interesting analyses from language guidance, showing feature maps on the Skewed-CelebA dataset, the effects of stylization, etc. These were pretty useful for understanding language guidance more deeply.

**Weaknesses:**

My main concern in the paper is related to novelty and clarity on its positioning. Given the large number of related papers in the field, I’m finding it a bit difficult to describe the guiding question the paper aims to answer. This leads to the weaknesses I describe below.

•	Motivation of the methods: Overall, I found the approach for ExLG and ImLG a bit difficult to motivate fully since I don’t see how they map language guidance approaches for vision representation learning papers from the past. I don’t see how these findings are interesting or relevant to the way people design language guidance for visual representation learning if ExLG or ImLG aren’t well motivated methods themselves. In particular, with ImLG, could the authors give some methods that use something similar?

•	Distinction with related papers: First, I strongly recommend that the authors write a related work section. This is necessary for positioning the paper in relation to other work that incorporates language guidance for visual representation learning. Overall, I found myself puzzled over the novelty of this paper. The paper finds many benefits from language guidance that has been found in prior papers that use language guidance [1, 2, 3]. The paper tries to differentiate itself from CLIP, which uses a joint language encoder by arguing that the approach uses a frozen language encoder. However, there are plenty of other papers that use a frozen text encoder [1, 2, 3] for language guidance. These papers also report similar findings of improvements over robustness, generalization, etc., although not all features are covered in the search I did.

•	Focus on vision domain: For a paper that has the title on language guided representation learning, I would have expected a focus on more domains than just vision. Would this extend to the other modalities? I would prefer if the title just stated that this was focused on vision instead.

Overall, in my opinion, the positioning and motivation of this paper need significant work. I would like to see a related work section added to the paper. If the authors can clarify their positioning in a satisfying manner, I may raise my score.

[1] El Banani et. al. Learning Visual Representations via Language Guided Sampling. CVPR 2023.
[2] Sariyildiz et. al. Learning Visual Representations with Caption Annotations. ECCV 2020.
[3] Stroud et. al. Learning Video Representations from Textual Web Supervision. arXiv 2021.

**Questions:**

•	Can the authors more clearly explain what ImLG is doing? I found myself confused about the approach.
•	I found it interesting to see cases where ImLG was worse than the baseline. For example, T4 in Figure 6. Would the authors mind providing some discussion?

---

> ### Author Response · Authors · 2024-11-21
> **Response to Reviewer W4mR**
>
> We appreciate the reviewer's acknowledgment of the paper's strengths and thank them for their detailed insights and feedback.
>
> - Firstly,  we acknowledge that the current title could imply a broader scope. Our focus in this paper is indeed specifically on language-guided visual representation learning. Our goal is to explore how language, as structured guidance, enriches vision-only feature representations
>
> - Novelty and Related works - Kindly see the Generic response above for more details.
> We revisit the foundational aspects of visual representation learning to examine how structured linguistic knowledge can influence and enhance vision encoders, addressing challenges such as robustness, shortcut learning, and continual learning.
>
> In response to the reviewer's suggestion, we have expanded the Related Works section in the paper. For instance, prior works like [1] use captions to find similar images for contrastive loss. Similarly, [2] rely on proxy tasks (e.g., predicting image tags from captions) for representation learning, which demands high-quality paired datasets and annotations. Approaches such as [3] utilize weak supervision by training video models on metadata embeddings. These methods explore multi-modal alignment, and function in joint embedding space and multi-modal and retrieval based tasks,  they rarely isolate the vision encoder’s learning dynamics and abilities.
>
> - Methods -
> The ExLG method provides explicit language embeddings as supervision. This approach is inspired by the human brain's utilization of language for higher-level abstractions and semantics, aligning with the concept of direct, verbalizable knowledge in cognitive systems. The impact of descriptions is also seen in vision-language alignment and image captioning tasks [3].
> The ImLG method, on the other hand, embeds language-derived contextual cues internally, bridging vision features with language context in a more implicit manner. Few works have employed frameworks where they project visual features to the input layer of LLMs directly [4,5,6]. [4] explore image feature embeddings as prompts to language models, and [5], which posits a information filtering hypothesis, that the language blocks in filtering and amplifying relevant visual features, ImLG ensures indirect supervision. We were also curious by the notion of "multi-modal neurons" [7], which respond to semantically related image and text embeddings, emphasizing a shared semantic abstraction.
>
> - ImLG - We wanted to project vision features to language encoder to evaluate if it amplifies the semantic information and improves the quality of representations. The implicit approach, while showing superior performance in challenging scenarios such as shortcut learning, texture bias analysis, and severe adversarial attacks, tends to underperform on IID (in-distribution) test accuracy compared to explicit methods. However, in IID settings, achieving optimal performance likely requires a more seamless alignment or harmony between the vision and language encoders during this transfer. We acknowledge this as an area for further exploration. The different guidance methods excel in distinct tasks, offering valuable insights into the nature of the supervision they provide.
>
> Additionally, please let us know if there is any specific information we can provide to enhance your support for our research.
>
>
> [4] Learning Visual Representations with Caption Annotations
> [5] Multimodal Few-Shot Learning with Frozen Language Models
> [6] Frozen Transformers in Language Models Are Effective Visual Encoder Layers
> [7]Multimodal neurons in pretrained text-only transformers.

---

> > ### Comment · Reviewer_W4mR · 2024-11-26
> >
> > I thank the authors for their response and I appreciate that they added a related work section to the paper. The authors state that their goal is to study the incorporation of language guidance into visual representation learning as far as leading to improvements in visual representation learning across many axes. They emphasize that they use a frozen encoder to create language representations and isolate changes in the visual encoder. At a first pass, this isn't necessarily very novel. The authors are applying a fairly narrow modification for multimodal contrastive representation learning and finding similar benefits as much of the prior work they cite. In my opinion, it's not entirely surprising even when the encoder is frozen and this makes the general conclusion that language guidance of visual representation learning can help unsurprising to the community. The main point of interest would then have to come from using methods that are well-motivated.
> >
> > After reading the authors’ response to my review, and the general response carefully to better understand the positioning of this paper, I have decided to keep my score. My primary reason for this is that I still believe the language guidance approaches, ExLG and ImLG, are not well motivated and this makes the audience and message of this paper unclear.

---

> > > ### Author Response · Authors · 2024-12-02
> > >
> > > Thank you for your feedback. We again wanted to summarize that our goal was to systematically explore the impact of structured language knowledge in supervised visual representation learning. While works focus on scaling vision-language models for zero-shot generalization or retrieval tasks, we diverge by targeting supervised setups with image-level pairs and discriminative classifier.  The papers on language supervision, focus on proxy tasks and sample-level captions using generative models. Our design choice stems from cognitive inspirations. Further, ExLG, also inspired by direct supervision from language, focuses on class-level descriptions to test the impact on different analyses. Additionally, curious by works that apply linear transformations between vision and language encoders to explore feature transfer, we explored the influence of ImLG. Our results highlight the distinctions and complementary strengths of these approaches
> > >
> > > Again quoting the example (from General response above) - for example, there has been no work trying to answer the research question: Can language impact specific vision challenges like shortcut learning and texture bias? Some approaches induce shape bias to address shortcut learning, but they remain constrained to pixel-based data without investigating language-guided solutions.
> > > By focusing on the specific supervised learning challenges often overlooked in scaling-focused literature, we provide insights that complement existing work, offering value to the community. We have provided additional summaries in the general response, along with more ablations and results. We kindly request the reviewer to review and see if they help clarify their concerns and offer support for our paper. Thank you for your time.

---

### Official Review · Reviewer_awyu · 2024-11-08

**Soundness:** 3
**Presentation:** 2
**Contribution:** 1
**Rating:** 3
**Confidence:** 3

**Summary:**

The paper focuses on the impact of using language model's representation on vision tasks. The paper uses two ways to use language model to "guide" the vision model, one is by explicitly align the vision model embedding and language model embedding, another is by implicitly use and freeze (part of) pretrained language model parameters as part of the model pipeline to make prediction on images. The authors demonstrated the usefulness of the two methods by showing that with language guidance, the model is more robust to out of distribution examples, texture bias, and adversarial attack, and it can do better on continual learning.

**Strengths:**

The paper covers experiments in extensive aspects to illustrate the benefits of language model guidance. The setting of the experiment is comprehensive, and I believe it can be reproduced.

**Weaknesses:**

- The idea behind this paper is not very novel. Starting from CLIP, it is well known that the alignment between language and vision can bring benefits (for example, [1] can do zero shot generalization to image classifications with new labels).
- The setting of the paper seems outdated. Nowadays, VLMs like LLaVA has been widely used, but the paper still focuses on ResNet and CIFAR-10. A good question here is what the implication of the results is, as the state-of-the-art models has already been using vision language alignment to achieve much more.
- The presentation of the paper can be improved. For example, eq (2) and (3). Seem like $f_v$ and $f_l$ are defined across a set of datapoints, $S_v(i, j)$ is the cosine similarity between image embeddings of two data points, indexed as $i$ and $j$. The current presentation is very misleading.
- Similarly, for Sec 4.2, the authors failed to clarify what is the input to classification head, and what is exactly the input to the language block. My concern here is that the paper uses ResNet-18, which is not natural to convert to input of language model block.
- Figure 3 is hard to read and interpret, thus the implication is unclear to me.

[1] Hanjie, Austin W., Ameet Deshpande, and Karthik Narasimhan. "Semantic supervision: Enabling generalization over output spaces." arXiv preprint arXiv:2202.13100 (2022).

**Questions:**

- In Figure 1, I don't understand why CKA can be applied here. What is $X$ and $Y$ in eq (4)?

---

> ### Author Response · Authors · 2024-11-21
> **Response to Reviewer awyu - 1**
>
> We would like to thank the reviewer for their feedback. Please find our responses below.
> Thank you for the feedback on presentation - In the revised version, we have refined the notation for improved readability.
>
> - Novelty - Our approach steps away from the mainstream goal of multi-modal training or developing parameter-efficient fine-tuning methods with pre-trained networks. Instead, we aim to revisit the fundamentals of visual representation learning by investigating how language, as structured knowledge, influences this process. Please also see the **Generic Response** and **Related Works**
>
> [1] also leverages pre-trained embeddings for target classes to do few-shot or zero-shot generalization. They train with image descriptions instead of class names, and hence also require to train a projection layer to map the dimensions. Further, during inference,  they predict the class corresponding to the class description with the highest softmax probability. They use BERT-small as the language encoder and ResNet18 as the image encoder, with inference reliant on descriptions.  While large Vision-Language Models (VLMs) like LLaVA excel in multimodal tasks, they face challenges such as reliance on large-scale multi-modal datasets, high computational costs, and vulnerability to catastrophic forgetting in sequential learning [3]. This motivates our decision to revisit foundational setups, allowing for a focused and controlled analysis of visual encoders' evolution under language guidance
>
> - Arch  -Additionally, we have included results with different architectures—ResNet50, VIT-Small—and evaluated on a larger dataset, ImageNet100, observing higher improvements as the model scales up.
>
> In our implicit method, the output embeddings from the ResNet encoder are passed through a linear transformation layer to align with the dimensionality required by the language model. This transformed embedding is then fed into the language model, which produces a refined representation that is sent to the classifier. The language model acts as a semantic filter, amplifying meaningful features and improving the overall representation quality. Similar approaches in prior works have demonstrated the utility of passing image encoder features as prompts to language models, with image encoders ranging from CNNs to transformers [1][2]. We also provide results with VIT vision encoder in the paper.
>
> - Figure 3. We provide here more explanation to the figure. We use Grad-CAM (Gradient-weighted Class Activation Mapping) to generate activation maps that visualize the regions of the image the models focus on while making predictions. Grad-CAM computes the gradients of the target class score with respect to the feature maps of the last convolutional layer, aggregates them to produce importance weights, and overlays a heatmap on the original image to indicate attention regions. Warmer colors (e.g., red) are areas of higher focus, while cooler colors (e.g., blue) show less attention. when applied to the Skewed-CelebA dataset, these maps reveal the contrast between the baseline model and the ExLG/ImLG models trained with language guidance. The baseline model predominantly focuses on spurious cues such as hair color, thus relying on superficial features correlated with the target labels. In contrast, the ExLG and ImLG models, guided by structured language semantics, focus on more salient and relevant facial features like eyes, mouth to make decision. This demonstrates that language guidance reduces shortcut learning by encouraging the models to form deeper, semantically meaningful representations, thereby addressing biases inherent in the dataset.
>
> - CKA - The alignment equation can be applied on any two feature vectors. In the first graph, we apply CKA between all images by passing them through an image encoder, extracting embeddings from the last block, and calculating pairwise CKA scores. For images, X and Y represent image feature descriptions, ex. X= real_f, Y = real_f - score of 1.0, next row, X=real_f, Y=clipart_f).
> For the CKA similarity between Images and text, X represents image features (real images passed through the image encoder), and Y represents text features (text descriptions passed through the text encoder, with embeddings extracted from the final layer. This analysis demonstrates that even with domain shifts, text descriptions provide a consistent semantic alignment, as reflected in the continual learning results (e.g., DN4IL in Table 3).
>
> [1] Linearly mapping from image to text space
> [2] Multimodal Few-Shot Learning with Frozen Language Models
> [3] Investigating the Catastrophic Forgetting in Multimodal Large Language Models

---

> > ### Comment · Reviewer_awyu · 2024-11-27
> > **About CKA**
> >
> > It seems you are just measuring the cosine similarities. What exactly is the image encoder and text encoder here? What type of training process have they been through?

---

> > > ### Author Response · Authors · 2024-11-28
> > >
> > > For the CKA analysis, we utilize a ResNet18 model pre-trained on ImageNet as the vision encoder and a Sentence Transformer language encoder trained on natural language sentences as the text counterpart. The purpose of applying CKA here is to measure the similarity between feature representations across domains, examining how these representations vary. In Figure 1, the object class remains consistent across domains, but the visual style (e.g., real, sketch, quickdraw) introduces domain variations that affect the learned representations. The textual descriptions, however, provide shared semantic concepts that remain invariant across domains and can serve as auxiliary supervision to improve classification accuracy across all domains. This sample are from the DN4Il dataset, that we use in continual learning experiments, where we see improvements in thh challenging domain incremental learning setting, where in each of sequential tasks, the domains keep changing.
> > > (Centered Kernel Alignment (CKA) is a used for measuring the similarity between two representations in neural networks (the formula is given in Section B in appendix). This was just an empirical analysis before designing our approach. Instead of learning a joint vision-and-language representation, we use linguistic conceptual knowledge to guide visual learning.

---

> ### Comment · Reviewer_awyu · 2024-11-28
>
> I understand what you are trying to convey here. However, the vision encoder and language encoder are trained without any multimodal data (so their outputting feature should not correlate with each other), but Figure 1 shows the similarities between quickdraw and the text is close to 1. It is easy to guess that the similarity between images and the text should all be small and similar, because the text feature can be treated as a random vector here (whether the text is "An airplane is ..." or "A car is ..." does not even matter). You probably used a different scaling for the similarity between images and text. In sum, all it says is just that there is no correlation between image and text features, which is obviously the case here.

---

> > ### Author Response · Authors · 2024-11-30
> >
> > Thanks for your response. In the case of text and image embeddings, as the scales were different, we had normalized during the CKA computation.
> > Below, we perform CKA only on the image data, and we use the models trained on the methods in our work. We test whether language context introduced by Explicit or Implicit Language Guidance enhances semantic alignment across domains.
> >
> > Base - ResNet18 trained on DN4IL dataset
> > ExLG - ResNet18 trained on the DN4IL dataset with descriptions for alignment (explicit guidance).
> > ImLG - ResNet18 trained on the DN4IL dataset with a Sentence Transformer on top of the vision encoder (implicit guidance).
> > The results are below (as we cannot submit the paper now, we could not show the plots, but we will update in the paper as well). ExLG shows improvement across domains, particularly in difficult domains such as painting and infograph. ImLG also improves but to a lesser extent.
> >
> > BASE
> > \
> > | Domain     | real | clipart | infograph | painting | sketch | quickdraw |
> > \
> > | real | 1.00 | 0.00 | 0.00 | 0.00 | 0.00 | 0.00 |
> > \
> > | clipart | 0.45 | 1.00 | 0.00 | 0.00 | 0.00 | 0.00 |
> > \
> > | infograph | 0.23 | 0.51 | 1.00 | 0.00 | 0.00 | 0.00 |
> > \
> > | painting | 0.31 | 0.40 | 0.19 | 1.00 | 0.00 | 0.00 |
> > \
> > | sketch | 0.19 | 0.55 | 0.46 | 0.16 | 1.00 | 0.00 |
> > \
> > | quickdraw | 0.25 | 0.73 | 0.51 | 0.14 | 0.77 | 1.00 |
> >
> >
> > ExLG
> > \
> > | Domain     | real | clipart | infograph | painting | sketch | quickdraw |
> > \
> > | real | 1.00 | 0.00 | 0.00 | 0.00 | 0.00 | 0.00 |
> > \
> > | clipart | 0.46 | 1.00 | 0.00 | 0.00 | 0.00 | 0.00 |
> > \
> > | infograph | 0.35 | 0.54 | 1.00 | 0.00 | 0.00 | 0.00 |
> > \
> > | painting | 0.35 | 0.64 | 0.27 | 1.00 | 0.00 | 0.00 |
> > \
> > | sketch | 0.26 | 0.54 | 0.42 | 0.31 | 1.00 | 0.00 |
> > \
> > | quickdraw | 0.32 | 0.72 | 0.54 | 0.32 | 0.79 | 1.00 |
> >
> >
> >
> > ImLG
> > \
> > | Domain     | real | clipart | infograph | painting | sketch | quickdraw |
> > \
> > | real | 1.00 | 0.00 | 0.00 | 0.00 | 0.00 | 0.00 |
> > \
> > | clipart | 0.43 | 1.00 | 0.00 | 0.00 | 0.00 | 0.00 |
> > \
> > | infograph | 0.41 | 0.51 | 1.00 | 0.00 | 0.00 | 0.00 |
> > \
> > | painting | 0.56 | 0.69 | 0.38 | 1.00 | 0.00 | 0.00 |
> > \
> > | sketch | 0.20 | 0.65 | 0.48 | 0.40 | 1.00 | 0.00 |
> > \
> > | quickdraw | 0.26 | 0.79 | 0.46 | 0.46 | 0.86 | 1.00 |

---

> > > ### Comment · Reviewer_awyu · 2024-12-01
> > >
> > > Thanks for the response, this sounds mildly related but cannot serve as a motivation. I suggest the authors think about the structure of the paper and organize the story carefully.
> > >
> > > As a side note, equations (2) and (3) are still ambiguous after revision. What is the definition of norm here? What does "feature matrices" mean?

---

> ### Author Response · Authors · 2024-12-01
>
> Thank you for the suggestion. To strengthen the narrative and flow, we reorder and clearly articulating our motivations and contributions. Our study was inspired by the following key questions: Can language be used to guide supervised vision learning? How can pre-trained language models produce rich representations in the visual domain to address challenges in supervised learning? Drawing inspiration from cognitive theories and prior work in language guidance in multi-modal tasks (Section D in appendix), we explored integrating language guidance into visual representation learning (with minimal overhead - class-level descriptions instead of image level in ExLG and using a light-weight language encoder in ImLG - new ablations to substantiate these are in Appendix section E.2, E.3 ) in various ways. We analyzed its impact on supervised learning challenges, Along with the gradcam activation map analyses to demonstrate the semantic richness of features learned through our approach and the ablations (in Appendix) we have also now added the CKA analysis at different layers.
>
> Loss - We reformulated the knowledge distillation loss (from teacher to student) for our setting. The objective is to ensure that inputs with similar semantic meanings (inferred from the language model) result in correspondingly similar activations in the vision encoder. We had added Section C in the appendix to include a more detailed explanation of the intuition behind the similarity-preserving alignment loss. We will clarify the equations more in the paper.
>
> f_v : activations from the vision encoder at a layer -  (We chose from last layer, just before the linear classifier
> \
>  f_v reshaped to (b, cxhxw) - batch size, channelxheightxwidth
> \
> f_t : activations from the language encoder, the encoded embeddings from sentence transformer
> \
> It is L2 normalization to obtain S_v and S_l.
> The similarity preserving Alignment loss ->  L_align = (S_v - S_l).pow(2).mean()

---

> > ### Author Response · Authors · 2024-12-02
> >
> > We again thank the reviewer for their feedback to help improve the quality of our work.
> > \
> > As a summary, our work focuses on supervised learning with image-label pairs - a discriminative classifier, exploring the under-researched area of leveraging language as structured knowledge to address challenges such as texture bias, shortcut learning, OOD generalization, robustness, and catastrophic forgetting. Through comprehensive ablations and analyses (detailed in the Appendix), we show the impact of different language guidances. By focusing on core supervised learning challenges often overlooked in scaling-focused literature and conducting extensive analyses, we aim to provide a holistic perspective and valuable insights. We have summarized the motivation and provided additional details about the new results in the general responses, and we kindly invite the reviewer to see if it helps address their concerns and increase support for our paper.

---

### Author Response · Authors · 2024-11-21
**General Response - 1**

We would like to thank all the reviewers for their time and effort in assessing our work and for providing insightful feedback. We have incorporated the suggested changes and submitted the revised paper, highlighting modifications in blue for clarity.

**Goal** - We revisit the fundamentals of visual representation learning and investigate how language, as structured knowledge, influences this process. A visual representation supervised learning setup (without contrastive loss or retrieval-based prediction), still face well-known challenges such as shortcut learning, texture bias, out-of-distribution (OOD) generalization, robustness, and catastrophic forgetting. Our aim was to evaluate the effectiveness of language guidance in addressing these critical challenges.

Models like CLIP, and other VLMs adopt contrastive learning and formulate classification as a retrieval task. They emphasize large-scale multi-modal learning for zero-shot generalization and multi-modal tasks, they rely on extensive datasets and computationally intensive training. These models also often face challenges in generalizing to images outside their pre-training datasets, requiring additional fine-tuning techniques. Further, recent works show that large multi-modal models are suffering from catastrophic forgetting, in continual learning setting. There are few language-guided techniques focusing on solving proxy tasks, employing generative image-captioning for pre-training, or using metadata for weak supervision. These methods operate within joint embedding spaces and rarely isolate or deeply analyze the vision encoder's properties.
(*Please see the new related works *Section D* for more details*)

We diverge from this paradigm by focusing on the basics and leveraging pre-trained language models to guide vision encoders without requiring multi-modal data or auxiliary tasks. Our two strategies, Explicit Language Guidance (ExLG) and Implicit Language Guidance (ImLG), draw inspiration from (1) *cognitive theories - System 1 and System 2 processing* , where System 2 involves deliberate and verbalizable knowledge (ExLG), and System 1 reflects the intuitive, indirect influence of ImLG; (2) *Global Workspace Theory (GWT)* supports the collaboration of explicit alignment and implicit modulation within a shared cognitive framework.

Further in ExLG, we adopt a knowledge distillation-inspired method using a similarity-preserving loss, which ensures that inputs with similar semantic meanings in the language model induce correspondingly similar activations in the vision encoder, thereby fostering a shared representation space. (*Section C in Appendix*)

Our goal is not to propose a state of the art method but to investigate and bring insights on how language can be leveraged for representational learning in vision models, and give a different perspective. Unlike literature that prioritize scaling, multi-modal data and IID performance, our analysis tests often-overlooked yet crucial aspects of robust visual representation learning. We also evaluate on the more challenging scenarios of class-incremental learning and domain-incremental learning, both of which are highly relevant, and are now being struggled by large pre-trained models.

See the table below for our analysis provided in the paper -


| Networks              | ResNet18           | ResNet50           | VIT                |
|------------------------|--------------------|--------------------|--------------------|
| Analysis              | Datasets           |                    |                    |
| IID                  | CIFAR10            | CIFAR100           | TinyImageNet, ImageNet100 |
| OOD                  | ImageNet-O         | ImageNet-R         | ImageNet-A        |
| Shortcut Learning    | Tinted-CIFAR10     | Skewed-CelebA      |                    |
| Texture Bias         | Stylized TinyImageNet |                  |                    |
| Adversarial Robustness | CIFAR10          |                    |                    |
| Continual Learning   | Seq-CIFAR10        | Seq-TinyImageNet   | DN4IL             |
|------------------------|--------------------|--------------------|--------------------|


**Modifications and Results in paper** :
Appendix -
* Detailed Related works and Motivation in section C in Related works
* Table 8 - Results on bigger vision encoder using ResNet50
* Table 9 - Results on a bigger dataset - ImageNet100, with ResNet18 and VIT image encoders
* Table 11 -
    * Results using simple and random language descriptions.
* Table 12, 13 and Figures 9,10,11,12
    * Impact of language model weights by experimenting with random weights and a CodeLM model trained on programming languages.

---

> ### Author Response · Authors · 2024-11-28
> **General Response - 2**
>
> **Summary**
>
> The task we target in this work is supervised learning in the vision domain, focusing specifically on datasets with image-label pairs. Our work explores an under-researched yet highly relevant area: leveraging language as structured knowledge to address persistent challenges in visual representation learning, including **shortcut learning, texture bias, out-of-distribution (OOD) generalization, robustness, and catastrophic forgetting**. We investigate whether and how language guidance can enhance robust representation learning, aiming to complement traditional one-hot encoding supervision efficiently.
>
> Inspired by cognitive theories we explore two types of supervision to improve the vision encoder’s learning process, bridging intuitive and semantic feature spaces. Our proposed methods, Explicit Language Guidance (ExLG) and Implicit Language Guidance (ImLG), rely on pre-trained language models without requiring multi-modal data or joint training. ExLG only needs *class-level descriptions* (e.g., for a dataset of 10k images and 100 classes, only 100 descriptions are required), while ImLG operates without any additional descriptions. Through  ablations, we demonstrate that even efficient, small-scale language models are sufficient for meaningful improvements.
>
> In contrast, vision-language models (VLMs) require joint training and operate in a contrastive setup for zero-shot tasks. Despite their scale, these models face challenges when tested on data different from their pre-training distribution and suffer from catastrophic forgetting in continual learning settings [1]. Existing language-guided methods typically rely on proxy tasks requiring additional annotations, infer captions during deployment, or need descriptions for every image, fail to isolate the vision encoder’s properties. For instance, there has been no work trying to answer the research question: Can language impact specific vision challenges like shortcut learning and texture bias? Some approaches induce shape bias to address shortcut learning [2], but they remain constrained to pixel-based data without investigating language-guided solutions.
>
> Our contributions include a systematic study of the impact of language guidance on visual representation challenges in a supervised setup. Through experiments and ablations, we explore the impact of language in various challenges by leveraging *small language models and class-level descriptions* to produce robust and semantically enriched visual representations.
> (This also further proven by activation maps in Figure 3 and 8, where the trained model learns to concentrate on relevant and semantic features after trained with language descriptions or language model).
>
>
> [1] Don't Stop Learning: Towards Continual Learning for the CLIP Model
> [2] ImageNet-trained CNNs are biased towards texture; increasing shape bias improves accuracy and robustness

---

> ### Author Response · Authors · 2024-12-02
> **General Response - 3**
>
> **Additional CKA analysis** - To further analyze the impact of language guidance, we performed a similarity analysis of feature embeddings for images belonging to the same class but from different domains. Similar to the setup in Figure 1, we selected images across domains where the model must learn shared features while minimizing domain-specific biases. As seen in the domain-incremental learning experiments within the continual learning setup, task performance often deteriorates.
> We analyzed the similarity of feature embeddings across multiple layers of our trained models using Centered Kernel Alignment (CKA). Since plots cannot be included here, we present the corresponding similarity matrix below (please see Figure 1 for reference). Our findings show that language guidance significantly improves cross-domain alignment at different layer of the ResNet18 architecture. This enhancement substantiates the role of language supervision in producing semantically richer and more robust visual representations.
>
>
> ResNet18 - block 3
> \
> BASE
> \
> | Domain | real | clipart | infograph | painting | sketch | quickdraw |
> \
> | real | 1.00 | 0.00 | 0.00 | 0.00 | 0.00 | 0.00 |
> \
> | clipart | 0.45 | 1.00 | 0.00 | 0.00 | 0.00 | 0.00 |
> \
> | infograph | 0.23 | 0.51 | 1.00 | 0.00 | 0.00 | 0.00 |
> \
> | painting | 0.31 | 0.40 | 0.19 | 1.00 | 0.00 | 0.00 |
> \
> | sketch | 0.19 | 0.55 | 0.46 | 0.16 | 1.00 | 0.00 |
> \
> | quickdraw | 0.25 | 0.73 | 0.51 | 0.14 | 0.77 | 1.00 |
>
> ExLG
> \
> | Domain | real | clipart | infograph | painting | sketch | quickdraw |
> \
> | real | 1.00 | 0.00 | 0.00 | 0.00 | 0.00 | 0.00 |
> \
> | clipart | 0.46 | 1.00 | 0.00 | 0.00 | 0.00 | 0.00 |
> \
> | infograph | 0.35 | 0.54 | 1.00 | 0.00 | 0.00 | 0.00 |
> \
> | painting | 0.35 | 0.64 | 0.27 | 1.00 | 0.00 | 0.00 |
> \
> | sketch | 0.26 | 0.54 | 0.42 | 0.31 | 1.00 | 0.00 |
> \
> | quickdraw | 0.32 | 0.72 | 0.54 | 0.32 | 0.79 | 1.00 |
> \
>
> ImLG
> \
> | Domain | real | clipart | infograph | painting | sketch | quickdraw |
> \
> | real | 1.00 | 0.00 | 0.00 | 0.00 | 0.00 | 0.00 |
> \
> | clipart | 0.43 | 1.00 | 0.00 | 0.00 | 0.00 | 0.00 |
> \
> | infograph | 0.41 | 0.51 | 1.00 | 0.00 | 0.00 | 0.00 |
> \
> | painting | 0.56 | 0.69 | 0.38 | 1.00 | 0.00 | 0.00 |
> \
> | sketch | 0.20 | 0.65 | 0.48 | 0.40 | 1.00 | 0.00 |
> \
> | quickdraw | 0.26 | 0.79 | 0.46 | 0.46 | 0.86 | 1.00 |
> \
>
> ResNet8 -block1
> \
> Base
> \
> | Domain     | real | clipart | infograph | painting | sketch | quickdraw |
> \
> | real | 1.00 | 0.00 | 0.00 | 0.00 | 0.00 | 0.00 |
> \
> | clipart | 0.03 | 1.00 | 0.00 | 0.00 | 0.00 | 0.00 |
> \
> | infograph | 0.02 | 0.13 | 1.00 | 0.00 | 0.00 | 0.00 |
> \
> | painting | 0.07 | 0.37 | 0.03 | 1.00 | 0.00 | 0.00 |
> \
> | sketch | 0.07 | 0.12 | 0.36 | 0.05 | 1.00 | 0.00 |
> \
> | quickdraw | 0.20 | 0.21 | 0.28 | 0.01 | 0.85 | 1.00 |
> \
>
> ExLG
> \
> | Domain     | real | clipart | infograph | painting | sketch | quickdraw |
> \
> | real | 1.00 | 0.00 | 0.00 | 0.00 | 0.00 | 0.00 |
> \
> | clipart | 0.04 | 1.00 | 0.00 | 0.00 | 0.00 | 0.00 |
> \
> | infograph | 0.04 | 0.20 | 1.00 | 0.00 | 0.00 | 0.00 |
> \
> | painting | 0.08 | 0.42 | 0.04 | 1.00 | 0.00 | 0.00 |
> \
> | sketch | 0.05 | 0.31 | 0.66 | 0.04 | 1.00 | 0.00 |
> \
> | quickdraw | 0.11 | 0.32 | 0.71 | 0.02 | 0.93 | 1.00 |
> \
>
> ImLG
> \
> | Domain     | real | clipart | infograph | painting | sketch | quickdraw |
> \
> | real | 1.00 | 0.00 | 0.00 | 0.00 | 0.00 | 0.00 |
> \
> | clipart | 0.01 | 1.00 | 0.00 | 0.00 | 0.00 | 0.00 |
> \
> | infograph | 0.03 | 0.16 | 1.00 | 0.00 | 0.00 | 0.00 |
> \
> | painting | 0.07 | 0.32 | 0.02 | 1.00 | 0.00 | 0.00 |
> \
> | sketch | 0.02 | 0.33 | 0.40 | 0.06 | 1.00 | 0.00 |
> \
> | quickdraw | 0.02 | 0.53 | 0.45 | 0.01 | 0.87 | 1.00 |
> \

---

### Meta-Review · Area_Chair_avTu · 2024-12-17

**Metareview:**

This paper received ratings of 5, 5, 5, 3, 3, and was unanimously recommended for rejection by all reviewers.
The papre investigates the role of natural language guidance in improving visual representation learning. The authors propose two approaches: Explicit Language Guidance for aligning vision and language embeddings using a similarity-preserving loss. And implicit Language Guidance for incorporating a frozen language model to implicitly enhance vision model training. The empirical study explores the effects of these approaches on various challenges. The results suggest that ExLG provides superior performance on most metrics, while ImLG shows advantages under certain conditions, such as robustness and shortcut learning.

Strengths:
- Interesting analyses: the authors present useful analyses, such as Grad-CAM activation maps and performance under texture bias.
- The use of cognitive inspirations (e.g., System 1 and System 2 processing) adds an interesting angle.

Area for improvements:
- The main critique from reviewers concerns the limited novelty of the approaches. Many findings are just to reiterate prior works (e.g., CLIP, contrastive vision-language models) that already show the benefits of aligning language and vision. The authors argue their focus is on supervised classification rather than contrastive training, but the distinction remains unclear to the reviewers.
- Ablations are insufficient. While the authors include additional ablations in response to reviewer feedback, critical questions remain, such as: does language guidance offer unique benefits over adding extra parameters or other forms of regularization? how do language models trained on non-semantic or random text compare systematically across tasks?
- Furthermore, the reviewers pointed out the presentation and clarity issues, such as the ambiguity persists in equations and experimental details (e.g., “stylization alpha,” "plasticity," and "stability").
- Motivation for ImLG remains to be weak, as it is unclear how the method advances beyond existing language-vision alignment approaches.
- Reviewers remain unconvinced of the paper’s unique contribution.

**Additional Comments On Reviewer Discussion:**

The rebuttal and discussion phases involved active engagement between the authors and reviewers, leading to clarification of several aspects of the manuscript while also underscoring some remaining concerns.

The authors addressed key points by incorporating additional ablation study and providing further clarifications. However, several issues raised by the reviewers persist. While reviewers acknowledged the improvements made, they found the methods insufficiently motivated and the claims largely incremental. Furthermore, concerns about robustness against confounding factors, such as differences in parameter counts, were not fully resolved.

One reviewer slightly increased their score but maintained the view that the work might be better suited for a specialized workshop rather than a flagship conference like ICLR. Reviewers also highlighted the need for more rigorous baseline comparisons to ensure fairness in evaluating the contributions. The authors are encouraged to incorporate the valuable feedback provided during the review and discussion phases to refine their work for future submissions.

---

### Decision · Program_Chairs · 2025-01-22

Reject